# Single-Cell Spatial Proteomics Clustering by Decoupling Spatiality and Expression

## Abstract

Single-cell spatial proteomics can reveal protein expression patterns while preserving the spatial structure of tissues, providing valuable insights into cellular functions and disease mechanisms. Spatial proteomics data clustering is a fundamental step in such studies, but it remains in the preliminary exploration phase, facing at least two prominent challenges: i) Functional regions within tissues often exhibit inherent area variations and *imbalanced* cell quantities, leading the model to favor features of majority classes, thus overshadowing the characteristics of minority ones. ii) Cellular identity is influenced by both intrinsic protein expression and the external spatial microenvironment; however, the *heterogeneity and potential conflicts* between these two information sources make it difficult to effectively identify subtle yet biologically significant cellular states. To overcome these issues, we propose a deep clustering framework named `spClust`. Our approach first introduces a spatially constrained synthetic minority oversampling technique to generate biologically meaningful cells of minority classes, alleviating the feature bias caused by cell type imbalance. Furthermore, we construct a spatiality adjacency graph and an expression similarity graph between cells, forming a decoupled dual-view contrastive learning architecture. We then define an adaptive mechanism to fuse the dual-view features and assign the soft cluster labels using dynamic prototypes, and further optimize labels by maximizing modularity. Extensive experiments on spatial proteomics datasets demonstrate that `spClust` effectively identifies minority cells and improves the distinction of different cells, confirming its effectiveness and superiority.

## 1 Introduction

As an important approach to unraveling the molecular mechanisms of life activities, proteomics has evolved from traditional bulk detection of tissue homogenates to the era of spatially resolved proteomics (Lundberg & Borner, 2019). Through techniques such as mass spectrometry imaging, spatial proteomics enables the simultaneous quantification of protein expressions and their spatial distributions within tissues (Rosenberger et al., 2023; Guo et al., 2021). This technique breakthrough provides an unprecedented perspective for revealing microenvironment heterogeneity, intercellular communication networks, and pathology (Kaufmann et al., 2022; Xu et al., 2024; Hong et al., 2025).

Clustering, a fundamental step in uncovering the intrinsic structure of spatial proteomics data (Nordmann et al., 2024; Li et al., 2024a; Karlsson et al., 2024; Unterauer et al., 2024), aims to cluster tissue regions or cells with similar expression patterns and spatial distribution characteristics. Well-clustered cells are beneficial for subsequent studies on cellular mechanisms, such as identifying differential protein markers, functionally homologous regions, or potential biological subgroups.

Spatial proteomics data present unique challenges for clustering algorithms, which must be carefully addressed to avoid misleading biological interpretations. One major issue arises from the inherent structural difference within tissues. Anatomically distinct regions, such as blood vessels, distal tubules, glomeruli, Interstitium, and proximal tubules in the kidney, exhibit substantial differences in physical size and cellular density. These natural variations cause a significant **imbalance in cell type distribution**. Conventional clustering methods often fail to accurately partition rare cell populations under these conditions, causing the omission of biologically important but infrequent cell states. The other issue is that the dual determinants of cellular identity should be accounted for

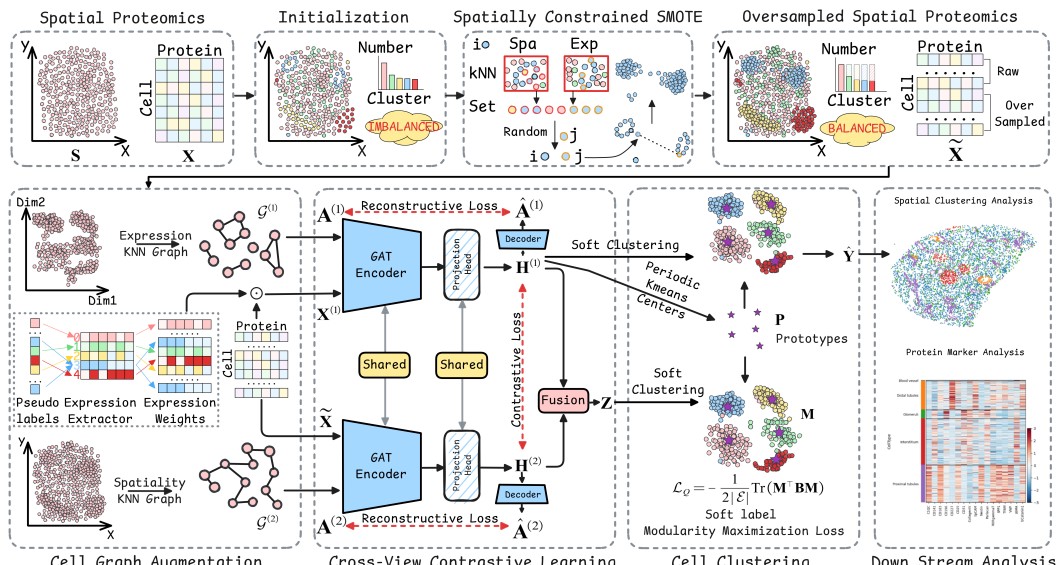

Figure 1: Schematic diagram of the proposed `spClust` framework. `spClust` firstly obtains pseudo labels by Leiden clustering and synthesizes biologically meaningful minority samples $\widetilde{\mathbf{X}}$ via a spatially constrained SMOTE oversampling to alleviate potential data imbalance. Next, it constructs an expression similarity graph $\mathcal{G}^{(1)}(\mathbf{X}^{(1)}, \mathbf{A}^{(1)})$ and a spatial adjacency graph $\mathcal{G}^{(2)}(\widetilde{\mathbf{X}}, \mathbf{A}^{(2)})$, and performs feature extraction on both graphs by a multi-head GAT encoder with shared weights to decouple the expression from spatial information. Then, it unifies the cross-view contrastive learning, graph reconstruction, and soft-label modularity maximization to optimize the clustering results, supporting spatial proteomics analysis, such as spatial clustering and protein marker discovery.

in accurate clustering. Proteomic expression profiles reflect the biochemical state of cells, while the spatial context reveals their functional role within tissue. These two modalities often exhibit **inherent heterogeneity and potential inconsistency**. Spatially adjacent cells may show distinct expression patterns, while similar cell types can be spatially dispersed. Without explicitly reconciling these complementary views, clustering algorithms struggle to distinguish subtle cellular states. Consequently, the development of an integrative computational framework that can jointly model molecular expression and spatial context is essential for advancing spatial proteomics analysis.

To address the above challenges, we propose `spClust` for single-cell spatial proteomics data, as illustrated in Figure 1. `spClust` leverages a spatially constrained oversampling mechanism with dual-view contrastive learning to rectify cellular distribution imbalance and learn consistent multi-view representations. Specifically, `spClust` introduces a spatially constrained synthetic minority oversampling technique (Spa-CSMOTE), which preserves both protein expression similarity and spatial neighborhood consistency when generating virtual cells. This process produces biologically plausible virtual cells, thereby mitigating clustering bias caused by class imbalance. On this basis, `spClust` constructs a local spatial adjacency graph and a global expression similarity graph to capture the heterogeneous relationships among cells from spatial and expression perspectives. By weight-shared graph attention encoders, it extracts cell features from both views and employs cross-view contrastive learning to encourage the model to learn consistent latent representations that align expression patterns with spatial contexts. `spClust` then defines an adaptive weighting mechanism to fuse the dual-view features and optimizes cluster assignments via dynamic prototype learning in an end-to-end manner. Experiments on multiple real-world spatial proteomics datasets demonstrate that `spClust` effectively identifies rare cell types and improves the quality of cell embeddings. Additionally, as an independent module, Spa-CSMOTE boosts the performance of conventional clustering methods on spatial proteomics data. Our main contributions are summarized as follows:

i) We propose a spatially constrained oversampling strategy, Spa-CSMOTE, that synthesizes virtual cells while maintaining the expression similarity and spatial adjacency, significantly reducing the clustering bias caused by imbalanced cell-type distributions.

ii) We develop a decoupled dual-view contrastive learning that can leverage spatial adjacency and expression similarity views to mitigate the heterogeneity between the two views effectively.

iii) We design an end-to-end deep clustering framework, spClust, which unifies spatial over-sampling, multi-view contrastive learning, and dynamic cluster assignment. The advantages of spClust are validated on multiple spatial proteomics datasets.

## 2 RELATED WORK

Single-cell spatial omics clustering (Lin et al., 2022; Yuan et al., 2024) necessitates the simultaneous examination of both molecular expression profiles and spatial locations of cells. Compared to traditional omics clustering, which relies solely on molecular expression similarity (Yin et al., 2025), single-cell spatial omics clustering faces more distinct analytical challenges. It must not only consider molecular expressions but also integrate spatial location information, focusing on the physical coordinates of samples and their neighborhood relationships. This means *avoiding misclassifications of spatially adjacent cells with subtle differences in expression, while also identifying special cells that are spatially separated yet share similar expression patterns*.

In the field of spatial transcriptomics, various clustering methods have been developed, providing powerful tools for analyzing the spatial transcriptional heterogeneity of tissues (Liu et al., 2024a). To fully leverage the multimodal data of spatial transcriptomics, SpaGCN (Hu et al., 2021) constructs a cell graph by integrating gene expression, spatial location, and histological images, and employs graph convolutional networks (GCN) to learn gene expression of cells and perform iterative clustering. Considering that spatially adjacent cells are not necessarily of the same type, STAGATE (Dong & Zhang, 2022) prunes predefined spatial neighborhoods using gene clustering results and utilizes a graph attention autoencoder to extract cell features for clustering. DeepST (Xu et al., 2022) utilizes a pre-trained model to extract tissue image features, combines them with gene expression and spatial location information, and employs autoencoders and denoising autoencoders to learn data representations. Spatial-MGCN (Wang et al., 2023) separately conducts graph convolution on graphs constructed with different information and fuses graph representations using an attention mechanism. stDGCC (Zhang et al., 2024) performs unsupervised feature extraction through contrastive learning by randomly augmenting the cell graph. More recently, SpatialLeiden (Müller-Bötticher et al., 2025) has been proposed to address the issue of Leiden's under-exploitation of spatial information.

In the field of spatial proteomics, scPROTEIN (Li et al., 2024b) dynamically performs cell graph attribute denoising and topological denoising while conducting contrastive learning to extract cell embeddings, and then uses Leiden clustering (Traag et al., 2019) to achieve cell clustering. However, scPROTEIN is only designed to learn cell embeddings, with a separation between embedding learning and downstream tasks that impairs clustering. scPROTEIN solely uses spatial information for cell graphs, neglecting connections between similar but distant cells, and the used random graph augmentation further reduces its interpretability.

As a key technology for analyzing tissue spatial heterogeneity at the protein level, spatial proteomics still lacks specialized clustering methods tailored to its data characteristics. Specifically, spatial proteomics data present a distinct challenge from transcriptomics. It exhibits more significant spatial heterogeneity, as it is directly influenced by processes such as post-translational modifications (Edwards et al., 2014) and intercellular communication (Quail & Walsh, 2024). Moreover, proteomics is lower-dimensional and denser, rendering methods designed for high-dimensional, sparse transcriptomics data susceptible to both overfitting and information loss. Critically, most existing methods neglect the underlying class imbalance inherent to cell populations in tissues. Proteins are the direct executors of life activities, and their spatial distribution is of great significance for understanding cellular functions and disease mechanisms. Therefore, developing clustering algorithms for spatial proteomics is an urgent need to advance life science research.

## 3 THE PROPOSED METHODOLOGY

**Notations** Spatial proteomics data typically include a protein expression matrix $\mathbf{X} \in \mathbb{R}^{n \times d}$ and a spatial coordinate matrix $\mathbf{S} \in \mathbb{R}^{n \times 2}$, where $n$ is the number of cells detected in a tissue slicing image, and $d$ is the number of detected proteins. However, current proteomics technologies are

limited by factors such as sensitivity, resulting in a far smaller number of proteins being detected compared to other omics. The Cartesian spatial coordinate of the $i$-th cell $\mathbf{x}_i \in \mathbb{R}^{1 \times d}$ in the tissue slicing is $(a_i, b_i)$, and each cell is labeled with a manually annotated tag $y_i$, with the number of cell types in a data section denoted as $c$. $\mathbf{A}$ is the adjacency matrix of the cell graph with self-loops, $|\mathcal{E}|$ is the number of edges, and $\mathbf{D}$ is the degree matrix. The pre-processing of spatial proteomics data is detailed in Section B.1 of the Appendix.

### 3.1 SPATIALLY CONSTRAINED SMOTE

The class imbalance problem (Guo et al., 2008; He & Garcia, 2009) has long been a significant challenge in data mining. Although techniques such as oversampling (Chawla et al., 2002; He et al., 2008) and undersampling (Liu et al., 2008) are widely used to address this issue, they still suffer from certain limitations in the field of spatial proteomics: i) *Dependence on ground-truth*. Addressing class imbalance assumes known class identities, which restricts their application in unsupervised settings. ii) *Neglect of spatial information*. Relying solely on expression profiles without considering spatial context can easily introduce biologically meaningless artifacts. We argue that a synthetic cell should ideally satisfy two conditions: having a protein expression profile similar to that of real minority classes, and being situated in a spatial microenvironment consistent with its phenotype. To meet these requirements, we propose a spatially constrained synthetic minority oversampling technique (Spa-CSMOTE). The core idea is to restrict the spatial coordinates of generated cells to biologically plausible local microenvironments.

As mentioned earlier, due to the absence of ground-truth labels, it is difficult to determine whether the data classes are balanced. As a result, we first employ the canonically used Leiden algorithm (Traag et al., 2019) in single-cell clustering to scrutinize preliminary clustering and obtain pseudo-labels $\widetilde{\mathbf{Y}}$, thereby identifying potential class imbalance in the data. If an imbalance is present, for each minority class $m$, we construct a candidate neighbor set $\mathcal{N}_m$:

$$\mathcal{N}_m = \mathcal{N}_m^{\text{spa}} \cup \mathcal{N}_m^{\text{exp}}, \tag{1}$$

where $\mathcal{N}_m^{\text{spa}}$ and $\mathcal{N}_m^{\text{exp}}$ denote the spatial $k$-nearest neighbors ($k$-NNs) and expression-based $k$-NNs of cells in the minority class $m$, respectively. For a random cell $i$ in class $m$, we randomly select a neighbor $j$ from $\mathcal{N}_m$ and generate a synthetic cell $\mathbf{x}'$ with protein expression given by:

$$\mathbf{x}' = \mathbf{x}_i + \lambda_1 * (\mathbf{x}_j - \mathbf{x}_i), \tag{2}$$

where $\lambda_1 \sim \text{Uniform}(0, 1)$ controls the interpolation ratio. For simplicity, we oversample all minority classes until they reach the size of the majority class. Similarly, we can generate the coordinate $\mathbf{s}'$ in the same way:

$$\mathbf{s}' = \mathbf{s}_i + \lambda_1 * (\mathbf{s}_j - \mathbf{s}_i). \tag{3}$$

However, the spatial positions of cells generated in this manner may lack biological plausibility, as the spatial microenvironment is not accounted. Therefore, we constrain the synthetic cells to the microenvironment of either cell $i$ or cell $j$, and propose the following generation scheme:

$$\mathbf{s}' = \mathbf{s}_i + \lambda_2 * (\mathbf{s}_j - \mathbf{s}_i), \tag{4}$$

where the probability of $\lambda_2$ between $[0, v_1)$ and $[1 - v_1, 1]$ is much greater than the probability between $[v_1, 1 - v_1)$. The default value of $v_1$ is 0.1, and the default probability is 0.9, meaning that the generated $\lambda_2$ has a 0.9 probability of falling within $[0, 0.1)$ and $[0.9, 1]$, with the two intervals having equal probabilities. The location of the synthetic cell is more likely to be constrained around $x_i$ or $x_j$ in this way. Thus, Spa-CSMOTE effectively addresses the issue of spatial inconsistency that may arise from completely random interpolation, ensuring that the synthetic data not only enhances class balance but also remains consistent with the real spatial tissue morphology. The balanced spatial proteomics data are denoted as $\widetilde{\mathbf{X}}$ and $\widetilde{\mathbf{S}}$.

### 3.2 CROSS-VIEW CONTRASTIVE LEARNING

Spatial location and protein expression are two inherent views of spatial proteomics data. However, significant view heterogeneity exists between two-dimensional physical spatial coordinates and multi-dimensional protein expression, making it challenging to effectively integrate them through

simple concatenation or early fusion. More importantly, although spatially adjacent and expression-similar cells often share the same functional phenotype, potential conflicts can arise between these two views. For instance, at tissue boundaries, cells in close spatial adjacency may belong to completely different types; conversely, cells with similar functional states may be spatially dispersed. Therefore, it is difficult to reconcile the complementarity and conflicts between these views to learn cell representations that simultaneously preserve spatial context consistency and semantic expression consistency. To address this, we propose a cross-view contrastive learning solution to decouple spatial and expression views. The structure of protein expression similarity $k$-NN graph $\mathcal{G}^{(1)}$ is constructed from two-dimensional embeddings obtained by a dimension reduction method (e.g., UMAP (McInnes et al., 2018)), with its adjacency matrix denoted as $\mathbf{A}^{(1)}$. The spatial adjacency $k$-NN graph $\mathcal{G}^{(2)}$ is based on spatial coordinates, represented by the adjacency matrix $\mathbf{A}^{(2)}$. These two graphs capture global similarity relationships and local microenvironment relationships, forming two natural augmentation views with different structures. For data augmentation, we design a learnable protein importance extractor to augment attributes of view one, avoiding the black-box feature of random augmentation:

$$\mathbf{X}^{(1)} = \mathbf{E}[\widetilde{\mathbf{Y}}, :] \odot \widetilde{\mathbf{X}}, \tag{5}$$

where $\mathbf{E} \in \mathbb{R}^{c \times d}$ represents the weights of the protein importance extractor, and $\widetilde{\mathbf{Y}}$ denotes the pseudo-labels of the cells. Subsequently, a shared-weights multi-head graph attention network (GAT) (Veličković et al., 2018) is employed to extract cell features, which are then mapped into the contrastive learning space via a shared-weights multi-layer perceptron (MLP) projection head:

$$\mathbf{Z}^{(1)} = \text{GAT}(\mathbf{X}^{(1)}, \mathbf{A}^{(1)}), \ \mathbf{Z}^{(2)} = \text{GAT}(\widetilde{\mathbf{X}}, \mathbf{A}^{(2)}), \tag{6}$$

$$\mathbf{H}^{(1)} = L_2(\text{Proj}(\mathbf{Z}^{(1)})), \ \mathbf{H}^{(2)} = L_2(\text{Proj}(\mathbf{Z}^{(2)})), \tag{7}$$

where $L_2$ denotes the $L_2$ normalization. The feature learning process of multi-head GAT is defined:

$$\mathbf{z}'_i = \|_{h=1}^{h'} \sigma\big(\sum_{j \in \mathcal{N}_i} \alpha_{ij}^h \mathbf{W}^h \mathbf{z}_j\big), \ \alpha_{ij} = \text{softmax}(\mathbf{a}^\top[\mathbf{W}\mathbf{z}_i \| \mathbf{W}\mathbf{z}_j]), \tag{8}$$

where $\|$ denotes the concatenation, $h'$ denotes the number of attention heads, $\sigma$ is the ReLU activation function, $\mathcal{N}_i$ is the neighbor set of node $i$, and $\mathbf{W}$ is the model weight matrix. $\mathbf{z}_i$ and $\mathbf{z}_j$ denotes the representation of node $i$ and $j$, respectively. The contrastive learning architecture follows the symmetric SimCLR (Chen et al., 2020):

$$\mathcal{L}_{\text{con}} = (\mathcal{L}_{\text{con}}^i + \mathcal{L}_{\text{con}}^j)/2, \tag{9}$$

$$\mathcal{L}_{\text{con}}^i = \frac{\exp(\text{sim}(\mathbf{h}_i^{(1)}, \mathbf{h}_i^{(2)})/\tau)}{\sum_{j=1}^n \mathbf{1}_{j \neq i} \exp(\text{sim}(\mathbf{h}_i^{(1)}, \mathbf{h}_j^{(2)})/\tau)}, \ \mathcal{L}_{\text{con}}^j = \frac{\exp(\text{sim}(\mathbf{h}_j^{(1)}, \mathbf{h}_j^{(2)})/\tau)}{\sum_{i=1}^n \mathbf{1}_{i \neq j} \exp(\text{sim}(\mathbf{h}_j^{(1)}, \mathbf{h}_i^{(2)})/\tau)}, \tag{10}$$

where $\text{sim}(\cdot, \cdot)$ denotes the cosine similarity. To further learn the structural information from the global expression graph and local spatial graph, the model reconstructs the adjacency matrices of both views through inner-product decoders. The reconstruction loss is defined as the cross-entropy between the normalized adjacency of the original graph and the reconstructed adjacency matrices:

$$\mathcal{L}_{\text{rec}} = \text{CE}(\hat{\mathbf{A}}^{(1)}, \mathbf{L}^{(1)}) + \text{CE}(\hat{\mathbf{A}}^{(2)}, \mathbf{L}^{(2)}), \tag{11}$$

where CE is the cross-entropy, $\mathbf{L} = \mathbf{D}^{-1/2} \mathbf{A} \mathbf{D}^{-1/2}$ is the normalized adjacency matrix.

Through the cross-view contrastive learning solution introduced above, the model aligns cellular spatial information with protein expression similarity in a unified representation space. This solution facilitates the learning of consistent representations that integrate both the external spatial context and the internal expression states of cells, providing a robust foundation for clustering.

## 3.3 SOFT-LABEL MODULARITY MAXIMIZATION BASED CLUSTERING OPTIMIZATION

End-to-end clustering allows the model to simultaneously learn discriminative representations and optimize cluster assignments, thereby enhancing the coherence and biological plausibility of the identified cellular communities. To fully exploit the spatial context-based representation and protein

expression semantic representation, dynamic weighting fusion is first designed to prevent simple addition from destroying their respective decoupled feature spaces:

$$\mathbf{Z} = \mathbf{F}[:, 0] * \mathbf{H}^{(1)} + \mathbf{F}[:, 1] * \mathbf{H}^{(2)}, \ \mathbf{F} = \text{softmax}((\mathbf{H}^{(1)} \| \mathbf{H}^{(2)})\mathbf{W}^{\top} + \mathbf{W}_b), \quad (12)$$

where $\mathbf{F} \in \mathbb{R}^{n \times 2}$ are dynamic fusion weights, $\mathbf{W} \in \mathbb{R}^{2 \times 2d'}$ is the weight matrix, and $\mathbf{W}_b \in \mathbb{R}^{1 \times 2}$ is the bias matrix.

The clustering centers are parameterized as prototypes denoted as $\mathbf{P} \in \mathbb{R}^{c \times d'}$, enabling joint optimization with other model components. To accelerate representation learning and avoid meaningless clustering, prototypes $\mathbf{P}$ are periodically updated every ten epochs using KMeans clustering centers derived from the expression-context embedding $\mathbf{Z}^{(1)}$. Based on the fused features and prototypes, soft clustering assignments are then computed as:

$$\mathbf{M} = \text{softmax}(\mathbf{Z} L_2(\mathbf{P})). \quad (13)$$

Inspired by the Leiden algorithm (Traag et al., 2019) commonly used in single-cell analysis, which uses a modularity maximization strategy (Newman, 2006) to enhance intra-cluster consistency and inter-cluster distinction, we introduce a clustering-specific objective based on modularity maximization to further unify representation learning and clustering in an end-to-end manner. To optimize the prototypes and embeddings simultaneously, we design a soft label modularity maximization loss:

$$\mathcal{L}_Q = -\frac{1}{2|\mathcal{E}|} \sum_{i,j} (\mathbf{A}_{i,j}^{(1)} - \frac{D_{i,i} D_{j,j}}{2|\mathcal{E}|})(\mathbf{M}_i^{\top} \mathbf{M}_j) = -\frac{1}{2|\mathcal{E}|} \text{Tr}(\mathbf{M}^{\top} \mathbf{B} \mathbf{M}), \quad (14)$$

where $\text{Tr}(\cdot)$ is the trace of a square matrix and $\mathbf{B}$ is the modularity. Compared with discrete hard label modularity (Yang et al., 2016; Liu et al., 2024b), soft label is continuous and more suitable for optimization through gradient descent.

To prevent the arbitrary augmentation of the protein expression extractor that may cause meaningless attributes or representation collapse, we design a Mean Squared Error (MSE) constraint:

$$\mathcal{L}_{\text{align}} = \text{MSE}(\mathbf{E}\mathbf{E}^{\top}, \mathbf{P}\mathbf{P}^{\top}). \quad (15)$$

The unified objective function of `spClust` is:

$$\mathcal{L} = \mathcal{L}_{\text{con}} + \gamma_1 * \mathcal{L}_{\text{rec}} + \gamma_2 * \mathcal{L}_Q + \gamma_3 * \mathcal{L}_{\text{align}}, \quad (16)$$

where $\gamma_1, \gamma_2, \gamma_3$ are parameterized learnable weights. The clustering results are obtained by:

$$\hat{y}_i = \arg \max_c \text{softmax}(\mathbf{H}^{(1)} L_2(\mathbf{P})). \quad (17)$$

Appendix A provides the computational complexity analysis and the pseudo-code of `spClust`.

## 4 EXPERIMENTS

### 4.1 DATA SOURCES AND METRICS

**Datasets** The DKD dataset (Kondo et al., 2024) comprises 17 kidney tissue CODEX images, all of which were collected from diabetic patients. This dataset quantifies the expression levels of 21 proteins using proteomic detection techniques. All cells in the samples have been manually annotated into six histological categories. In this study, three representative samples were selected for analysis, numbered "000", "004", and "013". The Breast cancer dataset (Danenberg et al., 2022) consists of 39-dimensional spatial proteomics data acquired using Imaging Mass Cytometry (IMC). In this study, we selected data slices numbered "6", "12", and "13". The melanoma dataset (Hoch et al., 2022) consists of 41-dimensional spatial proteomics data also acquired using IMC. In this study, we selected data slices numbered "103", "119", and "160". Please refer to Table B1 in the Appendix for more statistics of each dataset.

**Metrics** Since the dataset contains ground-truth labels, we employ the widely used clustering accuracy (ACC), Adjusted Rand Index (ARI), Normalized Mutual Information (NMI), and macro-F1 score (F1) to evaluate performance. The larger the value, the better the clustering performance.

Table 1: Comparison of clustering performance (ACC, ARI, NMI, and F1) for nine algorithms. Boldfaced and underscored values indicate the best and second-best result, respectively.

| Dataset | Metric | Base Methods | | | Transcriptomics Methods | | | | Spatial Proteomics Methods | |
|---|---|---|---|---|---|---|---|---|---|---|
| | | KMeans | Ward | Leiden Sci. Rep.'19 | scDC Nat. MI'19 | SpaGCN Nat. Met.'21 | stDGCC Bioinfo.'24 | SpatialLeiden Gen. Bio.'25 | scPROTEIN Nat. Met.'24 | Ours |
| **DKD_000** | ACC | .810±.000 | .814±.000 | .906±.001 | .789±.069 | .861±.038 | .760±.068 | .849±.022 | .880±.026 | **.914±.004** |
| | ARI | .727±.001 | .762±.000 | .804±.002 | .641±.113 | .769±.027 | .579±.088 | .713±.037 | .784±.017 | **.823±.009** |
| | NMI | .598±.001 | .624±.000 | **.747±.002** | .577±.078 | .681±.027 | .523±.038 | .718±.016 | .708±.032 | .740±.010 |
| | F1 | .661±.001 | .640±.000 | .856±.002 | .542±.091 | .767±.077 | .558±.062 | .720±.011 | .794±.063 | **.870±.007** |
| **DKD_004** | ACC | .716±.000 | .674±.000 | .785±.049 | .686±.032 | .729±.024 | .685±.071 | .791±.045 | .758±.046 | **.846±.011** |
| | ARI | .537±.000 | .540±.000 | .616±.030 | .530±.022 | .585±.027 | .430±.116 | .612±.039 | .591±.029 | **.664±.015** |
| | NMI | .529±.000 | .547±.000 | .614±.026 | .534±.021 | .585±.024 | .448±.069 | .607±.027 | .596±.021 | **.619±.010** |
| | F1 | .576±.000 | .556±.000 | .577±.042 | .539±.027 | .608±.029 | .529±.078 | .660±.018 | .595±.058 | **.664±.005** |
| **DKD_013** | ACC | .755±.000 | .770±.000 | .824±.018 | .757±.067 | .742±.059 | .725±.059 | .840±.029 | .799±.051 | **.850±.016** |
| | ARI | .567±.000 | .600±.000 | .638±.010 | .539±.078 | .524±.080 | .533±.074 | .627±.046 | .598±.063 | **.659±.009** |
| | NMI | .493±.000 | .512±.000 | .585±.019 | .523±.026 | .559±.030 | .459±.029 | **.633±.019** | .566±.020 | .620±.012 |
| | F1 | .581±.000 | .585±.000 | .645±.020 | .548±.056 | .621±.029 | .545±.046 | .693±.008 | .632±.022 | **.720±.038** |
| **Breast_6** | ACC | .768±.002 | .776±.000 | .760±.013 | .800±.076 | .582±.018 | .634±.056 | .609±.025 | .593±.022 | **.843±.034** |
| | ARI | .660±.002 | .642±.000 | .614±.063 | .632±.124 | .363±.007 | .185±.152 | .356±.004 | .363±.006 | **.675±.022** |
| | NMI | .512±.003 | .488±.000 | .478±.015 | .509±.035 | .443±.009 | .169±.064 | .431±.004 | .428±.012 | **.516±.022** |
| | F1 | .517±.002 | .641±.000 | .609±.026 | .670±.100 | .484±.009 | .415±.042 | .491±.009 | .489±.008 | **.726±.078** |
| **Breast_12** | ACC | .390±.009 | .411±.000 | .483±.008 | .369±.030 | .453±.043 | .457±.017 | .431±.029 | .328±.025 | **.499±.044** |
| | ARI | .192±.005 | .219±.000 | .282±.018 | .192±.025 | .268±.042 | .192±.015 | .225±.032 | .169±.015 | **.309±.061** |
| | NMI | .337±.003 | .318±.000 | .366±.010 | .357±.005 | **.401±.014** | .274±.007 | .344±.009 | .370±.008 | .333±.026 |
| | F1 | .318±.012 | .319±.000 | **.329±.006** | .285±.013 | .305±.025 | .308±.012 | .280±.012 | .241±.023 | .274±.024 |
| **Breast_13** | ACC | .754±.001 | .747±.000 | .747±.001 | .737±.010 | .786±.011 | .622±.084 | .781±.024 | .732±.071 | **.800±.012** |
| | ARI | .625±.001 | .610±.000 | .619±.000 | .610±.009 | .655±.011 | .403±.158 | .625±.013 | .602±.102 | **.662±.009** |
| | NMI | .529±.001 | .494±.000 | .524±.003 | .524±.011 | .546±.018 | .352±.116 | **.556±.023** | .551±.038 | .536±.007 |
| | F1 | .659±.001 | .647±.000 | .649±.000 | .658±.014 | **.718±.021** | .463±.074 | .598±.041 | .652±.070 | .714±.009 |
| **Melanoma_103** | ACC | .537±.040 | .599±.000 | .579±.026 | .360±.013 | .545±.068 | .642±.023 | .425±.031 | .475±.061 | **.668±.039** |
| | ARI | .348±.032 | .407±.000 | .421±.037 | .194±.016 | .365±.056 | .500±.024 | .246±.021 | .328±.081 | **.566±.066** |
| | NMI | .433±.010 | .401±.000 | .421±.035 | .181±.012 | .447±.020 | .413±.035 | .367±.024 | .283±.033 | **.484±.032** |
| | F1 | .297±.015 | .439±.000 | .314±.047 | .236±.009 | .406±.056 | .417±.044 | .303±.046 | .257±.039 | **.453±.070** |
| **Melanoma_119** | ACC | .465±.015 | .479±.000 | .489±.009 | .332±.008 | .514±.047 | .516±.025 | .470±.007 | .348±.003 | **.568±.041** |
| | ARI | .233±.008 | .220±.000 | .192±.028 | .100±.008 | .274±.040 | .245±.024 | .236±.007 | .117±.009 | **.284±.029** |
| | NMI | .316±.012 | .264±.000 | .254±.030 | .057±.006 | **.364±.020** | .267±.018 | .321±.008 | .169±.012 | .291±.020 |
| | F1 | .252±.009 | .267±.000 | .222±.017 | .155±.003 | **.355±.035** | .312±.027 | .298±.010 | .201±.012 | .273±.033 |
| **Melanoma_160** | ACC | .463±.016 | .475±.000 | .500±.011 | .288±.009 | .500±.020 | .456±.007 | .471±.009 | .398±.019 | **.512±.023** |
| | ARI | .254±.013 | .229±.000 | .229±.006 | .094±.007 | .300±.024 | .222±.011 | .285±.006 | .212±.016 | **.325±.013** |
| | NMI | .313±.009 | .294±.000 | .321±.007 | .068±.004 | .332±.019 | .261±.005 | .297±.008 | .228±.017 | **.345±.017** |
| | F1 | .311±.009 | .405±.000 | .350±.007 | .188±.009 | **.434±.037** | .344±.012 | .320±.027 | .297±.028 | .421±.040 |
| **Average Rank** | | 6 | 5 | 2 | 9 | 3 | 8 | 4 | 7 | **1** |

## 4.2 BASELINES

Three categories of methods serve as baselines: i) **base clustering methods** (KMeans (Hartigan & Wong, 1979), Ward hierarchical clustering (Ward Jr, 1963), Leiden (Traag et al., 2019)), which rely on feature similarity independent of spatial information; ii) **transcriptomics clustering methods** (stDC (Tian et al., 2019), SpaGCN Hu et al. (2021), stDGCC Zhang et al. (2024), SpatialLeiden (Müller-Bötticher et al., 2025)) that integrate gene expression and spatial information; and iii) **spatial proteomics method** (scPROTEIN Li et al. (2024b)).

## 4.3 IMPLEMENTATION DETAILS

The main experiments were conducted on a server with an Intel(R) Xeon(R) Platinum 8468V CPU (125GB) and an NVIDIA L40 GPU (48GB), while the calculation cost experiments were conducted on a personal computer with an Intel(R) Core(TM) i5-14400F CPU (32GB) and an NVIDIA GeForce RTX 5060 (8GB). The program was coded with Python 3.11.13 and PyTorch 2.5.1. For `spClust`, the number of attention heads was set to 2, and the dimension of each GAT layer was set to [64, 32, 16] for most datasets. For the projection head, the number of layers was set to 3. The GAT was pretrained via graph reconstruction loss with at least 50 epochs. Then, KMeans was conducted on the embedding learned by the pretrained GAT to warm up and initialize prototypes and pseudo-labels. Each dataset was trained for at least 100 epochs, and the learning rate was set to 1e-3 on most datasets. The number of neighbors in the global expression graph and the local spatial graph was set to 15. We conducted ten random repeats and recorded the mean and standard deviation of the results. Please refer to Section B.2 in the Appendix for more experimental settings about our `spClust` and baselines.

## 4.4 COMPARISON ANALYSIS

The clustering results on nine datasets are listed in Table 1. We have the following observations:
i) Our `spClust` achieves the best ACC and ARI results on all datasets, attaining the highest overall

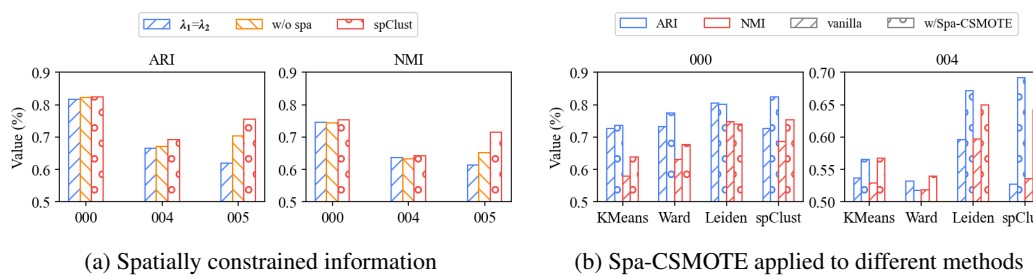

Figure 2: Ablation results on Spa-CSMOTE.

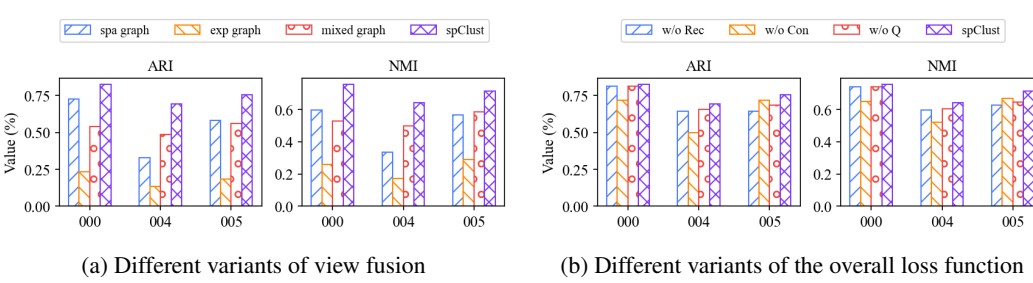

Figure 3: Ablation results on view decoupling and loss function.

ranking. These results show that spClust can effectively cluster spatial proteomics data. However, since spClust relies on an unsupervised data balancing strategy, the reliability of oversampled data may affect the final representation learning and clustering performance.

ii) Although conventional methods such as KMeans, Ward, and Leiden do not employ deep learning for feature extraction, they still achieve competitive clustering results. Notably, the Leiden algorithm constructs a similarity graph of protein expression and maximizes community modularity for clustering. This suggests that using protein expression alone may be effective for spatial proteomics clustering, and the modularity maximization can produce cohesive and well-separated clusters.

iii) Spatial transcriptomics methods can also be applied to tabular-form spatial proteomics data. However, since current spatial transcriptomics profiling typically targets high-dimensional sparse data, their pre-processing and feature learning strategy may not be directly suitable for contemporary spatial proteomics data with low dimensionality and high density. As a result, they are significantly lost to spClust.

iv) spClust, SpaGCN, SpatialLeiden, and Leiden consider the spatial information or graph structures, generally outperforming rivals that disregard this information. This suggests that spatial or structural relationships among cells contain valuable information for clustering, as the follow-up ablation experiments will further prove.

## 4.5 ABLATION STUDY

To validate the effectiveness of our spClust, we conducted comprehensive ablation studies.

For Spa-CSMOTE, we designed two variants: "$\lambda_1 = \lambda_2$" indicates that the locations of synthetic cells are randomly generated along the line connecting the base cell and its neighbor, without spatial microenvironment constraints; "w/o spa" denotes that only expression-similar cells are considered when generating synthetic cells, without incorporating spatially adjacent cells. The results in Figure 2a demonstrate that spatial microenvironment constraints can guide the synthetic cell placement, and constructing the candidate pool using both similar and neighboring cells is effective. Furthermore, as shown in Figure 2b, Spa-CSMOTE can serve as a plug-and-play module that enhances the performance of conventional clustering methods, suggesting that data imbalance indeed compromises the effectiveness of these clustering algorithms.

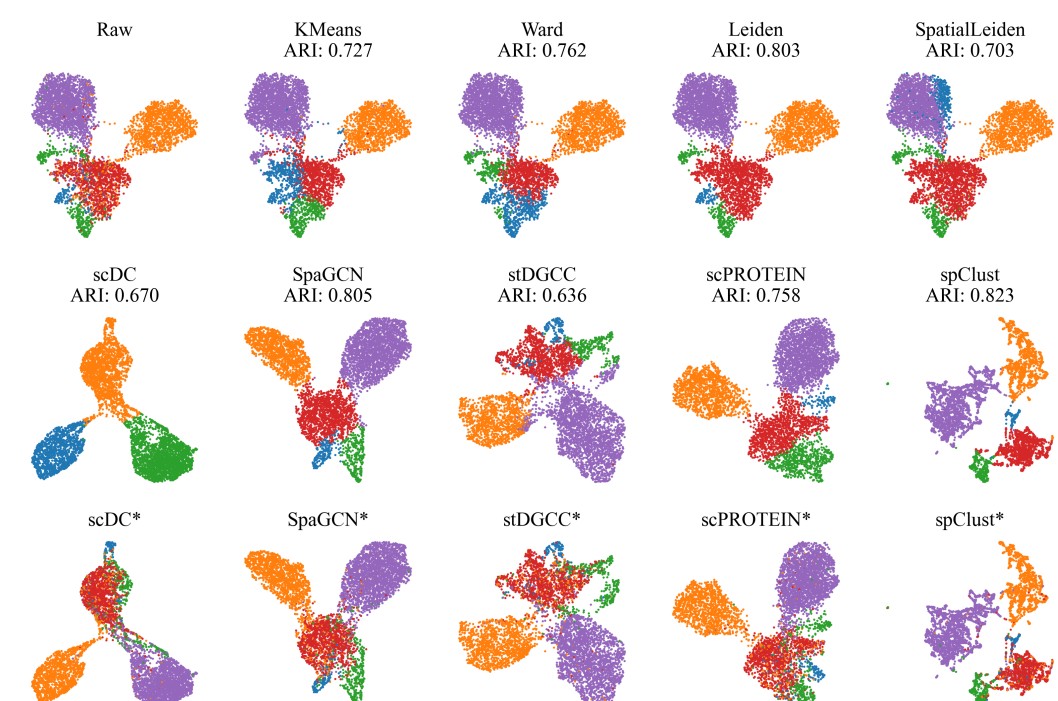

Figure 4: Visualization of cell embedding via 2D UMAP dimensionality reduction on the dataset "DKD_000". The first figure is the raw expression labeled with the ground truth. The top row displays the clustering results of methods that do not use representation learning. The second row displays the embedding visualization of the representation learning-based methods, with labels indicating the corresponding cluster labels. The third row presents the same embeddings, but with labels replaced by ground-truth labels, and method names are marked with an asterisk (*) for distinction.

For the expression–spatial dual-view decoupling module, we designed three variants: "spa graph" refers to constructing the cell graph using only spatial information by random augmentation to generate two views. Similarly, "exp graph" indicates using only protein expression to build the graph with random augmentations, while "mixed graph" fuses the spatial adjacency $k$-NN and expression similarity $k$-NN graphs, again with random augmentations applied. The results in Figure 3a show that the decoupling module in `spClust` is more suitable for cross-view contrastive learning. The reduced clustering performance of the mixed graph suggests potential conflicts between spatiality and expression.

Additionally, to evaluate the contribution of each loss function in `spClust`, we ablated the reconstruction loss ("w/o Rec"), the contrastive loss ("w/o Con"), and the modularity maximization loss ("w/o Q"), respectively. The results in Figure 3b confirm that the removal of each loss leads to degraded clustering performance, especially contrastive learning loss. Contrastive learning loss is the key to linking the decoupled spatial embeddings and expression embeddings of cells. Aligning the two in the contrastive learning space enables cells to learn the essential features of spatiality and expression.

## 4.6 VISUALIZATION ANALYSIS

The 2D UMAP visualizations of raw expressions and embeddings learned by different methods are depicted in Figure 4. It can be observed that all methods can identify the major class of cells, while KMeans, Ward, SpatialLeiden, and scDC struggle to recognize the minor class of cells accurately. Although other methods can detect the minority, their features have low discriminability from those of the majority. In contrast, our `spClust` not only achieves accurate identification of the minority but also ensures that the features of each class exhibit both strong discriminability and high cohesion. This advantage stems from two aspects. First, Spa-CSMOTE effectively balances the sample size of

Table 2: Clustering performance of `spClust` with different Initialized Pseudo-Labels on the DKD_000 Dataset.

| Metric | Stage | Leiden with Different Seeds | | | Different Initialization Methods | | | Extreme Cases | | STDEV |
|---|---|---|---|---|---|---|---|---|---|---|
| | | 0 | 42 | 2026 | SpatialLeiden | KMeans | Ward | Random | Ground-Truth | |
| ACC | Initial | 0.906 | 0.905 | 0.907 | 0.715 | 0.742 | 0.814 | 0.209 | **1.000** | 0.247 |
| | `spClust` | **0.912** | **0.912** | **0.913** | **0.905** | **0.895** | **0.849** | **0.687** | 0.907 | **0.078** |
| ARI | Initial | 0.804 | 0.801 | 0.806 | 0.483 | 0.654 | 0.762 | 0.000 | **1.000** | 0.306 |
| | `spClust` | **0.820** | **0.821** | **0.822** | **0.819** | **0.807** | **0.789** | **0.499** | 0.821 | **0.112** |
| NMI | Initial | **0.746** | 0.744 | **0.748** | 0.563 | 0.582 | 0.624 | 0.001 | **1.000** | 0.288 |
| | `spClust` | 0.744 | **0.746** | 0.746 | **0.729** | **0.716** | **0.690** | **0.600** | 0.735 | **0.050** |
| F1 | Initial | 0.856 | 0.855 | 0.856 | 0.718 | 0.565 | 0.640 | 0.177 | **1.000** | 0.256 |
| | `spClust` | **0.868** | **0.866** | **0.866** | **0.838** | **0.812** | **0.747** | **0.477** | 0.849 | **0.133** |

different classes. Second, the cross-view contrastive learning with expression-spatiality decoupling successfully captures both the intrinsic expression features and extrinsic spatiality of cells.

### 4.7 ROBUSTNESS ANALYSIS

Intuitively, the initial clustering pseudo-labels of `spClust` affect the judgment of class imbalance and the generation of over-sampled cells, thereby influencing model robustness. We verified the robustness of `spClust` against initial pseudo-labels of varying quality on the DKD_000 dataset, including: Leiden clustering pseudo-labels generated with different random seeds (0, 42, 2026), pseudo-labels from distinct clustering methods (Leiden, SpatialLeiden, KMeans, Ward), and two extreme cases (random labels and ground-truth labels). As shown in Table 2, `spClust` outperforms the initial clustering results under all types of pseudo-labels, with enhanced stability, except for the ground-truth. Even when using completely random labels, the model can still perform effectively. Notably, using fully matched

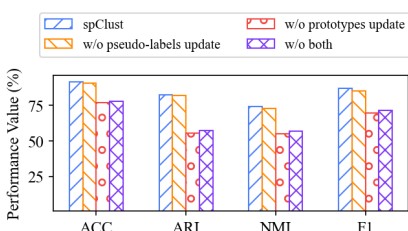

Figure 5: Clustering performance of different pseudo-labels and prototypes update strategies.

ground-truth labels does not yield perfect clustering results. This is because the construction of the expression similarity graph and the spatial adjacency graph does not utilize label information; incorrect neighbor connections lead to feature smoothing, which decreases the discriminability of clusters. In fact, ground-truth labels may not reflect the intrinsic distribution of the data, whereas the pseudo-labels used in our experiments capture the true similarity distribution, highlighting the data-driven rather than label-driven design philosophy of our method. Furthermore, the results in Figure 5 demonstrate that the periodic update mechanism for prototypes and pseudo-labels in `spClust` promptly rectifies the effects of erroneous pseudo-labels and prototypes, further confirming the robustness. Appendix D.3 provides additional robustness results for `spClust` with different initial pseudo-labels.

## 5 CONCLUSION

In this study, we propose an end-to-end deep clustering framework, `spClust`, for single-cell spatial proteomics clustering. `spClust` generates biologically meaningful minority cells using spatially constrained synthetic minority oversampling technology to balance cell counts across classes, effectively alleviating the problem where minority-class features are overshadowed by majority-class features in unsupervised learning. Meanwhile, verified by conventional clustering algorithms, this oversampling module demonstrates the plug-and-play capability to enhance clustering performance. Additionally, it defines cross-view contrastive learning with spatial-expression decoupling to effectively characterize both the extrinsic spatial microenvironment features and the intrinsic protein expression features of cells, thereby further improving the quality of cell embedding and clustering. Future research will focus on more reliable and flexible unsupervised imbalanced learning techniques for optimizing the quality of synthetic data.

## 6 ETHICS STATEMENT

We promise that we have read the ICLR Code of Ethics, and this article has not raised any questions regarding the Code of Ethics.

## 7 REPRODUCIBILITY STATEMENT

We promise that `spClust` is reproducible. The code is available at `https://anonymous.4open.science/r/spClust-6842`.

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

## A  PSEUDO-CODE OF ALGORITHM AND COMPLEXITY ANALYSIS

---

**Algorithm 1** `spClust`: Single-cell Spatial Proteomics Clustering by Decoupling Spatiality and Expression

---

**Input**: Spatial proteomics data with expression $\mathbf{X}$ and coordinate $\mathbf{S}$
**Parameter**: Temperature parameters $\tau$, iterations $M_1$, clusters $c$, number of nearest neighbors $k$
**Output**: Clustering results $\hat{\mathbf{Y}}$

1: Pre-process data.
2: Initialize class pseudo-labels $\widetilde{\mathbf{Y}}$ via Leiden clustering.
3: Oversample via Spa-CSMOTE to generate balanced data.
4: Construct cell graphs $\mathcal{G}^{(1)}$ and $\mathcal{G}^{(2)}$.
5: Pretrain GAT with balanced data via graph reconstruction loss.
6: Update prototypes and pseudo-labels via KMeans on feature learned by pretrained GAT.
7: **for** $l \longrightarrow 1, \cdots, M_1$ **do**
8:    Augment attributes of $\mathcal{G}^{(1)}$ via Eq. (5).
9:    Learn projected cell representations $\mathbf{H}^{(1)}$ and $\mathbf{H}^{(2)}$ via Eq. (7).
10:    Obtain fused embedding from $\mathbf{H}^{(1)}$ and $\mathbf{H}^{(2)}$ via Eq. (12).
11:    Calculate soft clustering assignment matrix $\mathbf{M}$ via Eq. (13).
12:    Update the unified loss $\mathcal{L}$ via Eq. (16).
13:    **if** $(l \% 10) == 0$ **then**
14:       Update prototypes $\mathbf{P}$ and pseudo-labels with KMeans centroids on $\mathbf{H}^{(1)}$.
15:    **end if**
16: **end for**
17: Calculate clustering results via Eq. (17).
18: **return** Clustering results $\hat{\mathbf{Y}}$.

---

`spClust` consists of multiple sequential steps, and its computational complexity can be analyzed step-by-step following the algorithm's execution workflow: Clustering labels are first initialized using the Leiden algorithm, with a computational complexity of $\mathcal{O}(n)$ for this step. Subsequently, Spa-CSMOTE is executed based on the obtained clustering pseudo-labels to generate balanced data. Assuming the number of cells in each minority class is $m_i$, the computational complexity of this step is $\mathcal{O}(\sum_{i=1}^{c-1} m_i(n + m_i d))$. Next, the processes of constructing the KNN graph and extracting features via GAT are carried out sequentially, with computational complexities of $\mathcal{O}(n^2)$ and $\mathcal{O}(n \sum_{i=1}^{3} d_{i-1}d_i + |\mathcal{E}| \sum_{i=1}^{3} d_i)$ respectively. Finally, the computational complexity of the loss function is collectively summarized as $\mathcal{O}(n^2)$. In addition, the auxiliary components involved in the workflow, such as KMeans clustering and UMAP dimensionality reduction, have a combined computational complexity of $\mathcal{O}(ncd)$.

Table A1: Time and space costs on the DKD_000 dataset.

| Metric | Shallow Methods | | | | Deep Methods | | | | |
|---|---|---|---|---|---|---|---|---|---|
| | **KMeans** | **Ward** | **Leiden** | **SpatialLeiden** | **scDC** | **SpaGCN** | **stDGCC** | **scPROTEIN** | **Ours** |
| **Time (s)** | 0.14 | 0.55 | 1.59 | 9.25 | 18.12 | 6.86 | 84.28 | 8.26 | 93.07 |
| **Peak CPU (MB)** | 303 | 273 | 392 | 919 | 1608 | 1441 | 1743 | 1783 | 9882 |
| **Peak GPU (MB)** | NA | NA | NA | NA | 43 | NA | 219 | 843 | 4869 |

For the DKD_000 dataset (comprising 5300 cells), Table A1 summarizes the comparative analysis of computational time and resource utilization: shallow methods (e.g., KMeans, Ward, and Leiden) offer high computational efficiency and low memory footprint, enabling efficient execution on a standard CPU. In contrast, deep learning-based approaches, including stDGCC and our `spClust`, demand greater computational and storage overhead. `spClust` encounters scalability limitations when handling large-scale datasets, constrained by its $\mathcal{O}(n^2)$ computational and spatial complexity.

Table B1: Statistics of nine spatial proteomics datasets after pre-processing.

| Dataset | DKD | | | Breast | | | Melanoma | | |
|---|---|---|---|---|---|---|---|---|---|
| | **000** | **004** | **013** | **6** | **12** | **12** | **103** | **119** | **160** |
| **#Cell** | 5387 | 10004 | 8573 | 871 | 3070 | 1029 | 7335 | 7809 | 2819 |
| **#Protein** | 21 | 21 | 21 | 39 | 39 | 39 | 41 | 41 | 41 |
| **#Class** | 5 | 5 | 5 | 3 | 10 | 4 | 7 | 9 | 8 |
| **#Max** | 2237 | 3787 | 2990 | 556 | 1724 | 490 | 3715 | 3817 | 756 |
| **#Min** | 129 | 57 | 203 | 61 | 55 | 117 | 206 | 96 | 86 |

# B  EXPERIMENTAL DETAILS

## B.1  PRE-PROCESSING

The statistics of nine datasets presented in Table B1 demonstrate the class imbalance challenge. Considering the "Other" or "unknown" category and the extremely minor category (¡50 cells) may be a potential source of noise, we eliminated these data. In accordance with the pre-processing protocols widely adopted in bioinformatics analysis Zhang et al. (2024), we first normalize the single-cell data by setting the target sum to 1 (except for Melanoma "160", which uses 1e6). Subsequently, we applied log1p to eliminate differences in protein expression magnitudes. Finally, we performed standardization to center the mean at zero (except for Melanoma "160") and truncated values greater than 10. The pre-processed data provides a better input basis for feature extraction.

## B.2  SETTINGS

For all methods, the random seed was initialized to 42 and incremented by one for each run to ensure reproducibility. The number of neighbors for all $k$-NN-based algorithms was set to 15, unless specified otherwise. The specific hyperparameter settings for the baseline methods were as follows:
**KMeans**: The n_init parameter was set to 20.
**Leiden**: The resolution parameter was determined using a binary search within the range of [1e-6, 3] to match the ground-truth number of clusters.
**SpatialLeiden**: The spatial neighborhood graph was constructed using Delaunay triangulation with 25 neighbors. The layer_ratio was set to 1.8, and its built-in algorithm was used to search for the optimal resolution.
**scDC**: The encoder dimensions were set to [256, 64, 32], with the random noise coefficient sigma at 2.5 and the clustering loss weight gamma at 1. The tolerance and learning rate were set to (0.01, 1.0), (0.001, 0.001), (0.001, 0.1) for DKD, Breast, and Melanoma, respectively.
**SpaGCN**: SpaGCN was run with its default recommended parameters.
**stDGCC**: The lambda_I parameter was set to 1.
**scPROTEIN**: The embedding and projection head dimensions were set to 128 and 64, respectively. The model was trained for 100 epochs with a learning rate of 1e-3. The train epoch and learning rate were set to (50, 1e-4), (50, 1e-3), (30, 1e-3) for DKD, Breast, and Melanoma, respectively.

The learnable parameters in Eq. (16) were initialized as 10, 10, and 0.01, respectively. The other parameters of spClust are detailed in Table B2.

Table B2: Parameters of spClust on nine spatial proteomics datasets.

| Dataset | norm_total_sum | zero_center | initialiation | Pretrain Epoch | Pretrain lr | Train Epoch | Train lr | hidden dims | temperature | knn | leiden neighbors | IR |
|---|---|---|---|---|---|---|---|---|---|---|---|---|
| DKD_000 | 1 | TRUE | Leiden | 100 | 1e-3 | 500 | 1e-3 | [64, 32, 16] | 0.5 | 15 | 25 | 1 |
| DKD_004 | 1 | TRUE | Leiden | 100 | 1e-3 | 200 | 2e-3 | [64, 32, 16] | 0.2 | 15 | 15 | 1 |
| DKD_013 | 1 | TRUE | Leiden | 50 | 1e-2 | 600 | 1e-3 | [64, 32, 16] | 0.8 | 15 | 25 | 1 |
| Breast_6 | 1 | TRUE | KMeans | 50 | 1e-2 | 300 | 1e-3 | [1024, 256, 16] | 0.1 | 10 | 25 | 1 |
| Breast_12 | 1 | TRUE | Leiden | 50 | 1e-3 | 500 | 1e-3 | [64, 32, 16] | 0.1 | 15 | 15 | 1 |
| Breast_13 | 1 | TRUE | KMeans | 50 | 1e-4 | 300 | 1e-3 | [64, 32, 16] | 1 | 15 | 15 | 1 |
| Melanoma_103 | 1 | TRUE | KMeans | 100 | 1e-3 | 500 | 1e-3 | [128, 32, 16] | 0.07 | 15 | 25 | 1 |
| Melanoma_119 | 1e6 | TRUE | KMeans | 100 | 1e-3 | 250 | 1e-3 | [64, 32, 16] | 0.005 | 15 | 25 | 2 |
| Melanoma_160 | 1 | FALSE | KMeans | 50 | 1e-3 | 500 | 1e-3 | [128, 32, 16] | 0.1 | 15 | 15 | 3 |

### B.3 METRICS

Clustering Accuracy (ACC) ranges from [0, 1] and reflects the proportion of correctly clustered samples. The Hungarian algorithm is used to match clustering labels with ground-truth labels before calculation. The Macro F1 score, also ranging from [0, 1], measures the model's ability to recognize all classes. It first computes the harmonic mean of precision and recall for each class, then takes the arithmetic mean of all class-specific F1 scores. The Adjusted Rand Index (ARI) adjusts the original Rand Index by subtracting random consistency, with values ranging from $[-1, 1]$: a value of one indicates perfect consistency, zero corresponds to random performance, and negative values indicate performance inferior to random. The Normalized Mutual Information (NMI) is based on mutual information normalized by entropy, with values in the range $[0, 1]$. A value closer to 1 indicates a higher degree of matching between the clustering results and the ground-truth labels. Notably, although balanced spatial proteomics data are used to learn cell embeddings, we only evaluate the original imbalanced data.

## C VISUALIZATION ANALYSIS OF SPA-CSMOTE

To intuitively compare the performance of spatial position-constrained (i.e., Eq. 3) and random-position (i.e., Eq. 4) cell synthesis strategies, we visualized the synthetic balanced dataset on DKD_kidney_000. Notably, randomly generated cells of Class 4 form a distinct banded distribution, which are prone to becoming neighboring cells of other classes, resulting in poor representation learning and degraded clustering performance. In contrast, spatially constrained synthesized cells cluster within their respective classes, exhibiting a more reasonable spatial distribution.

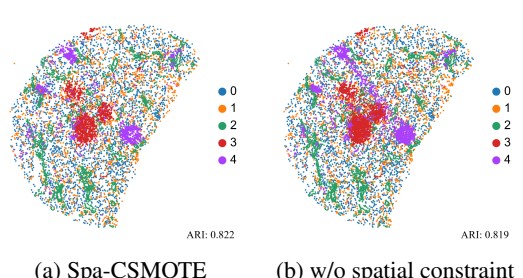

(a) Spa-CSMOTE     (b) w/o spatial constraint

Figure C1: Comparison of synthetic cells visualization under different spatial constraints.

## D ADDITIONAL EXPERIMENTAL RESULTS

### D.1 SPATIAL CLUSTERING VISUALIZATION

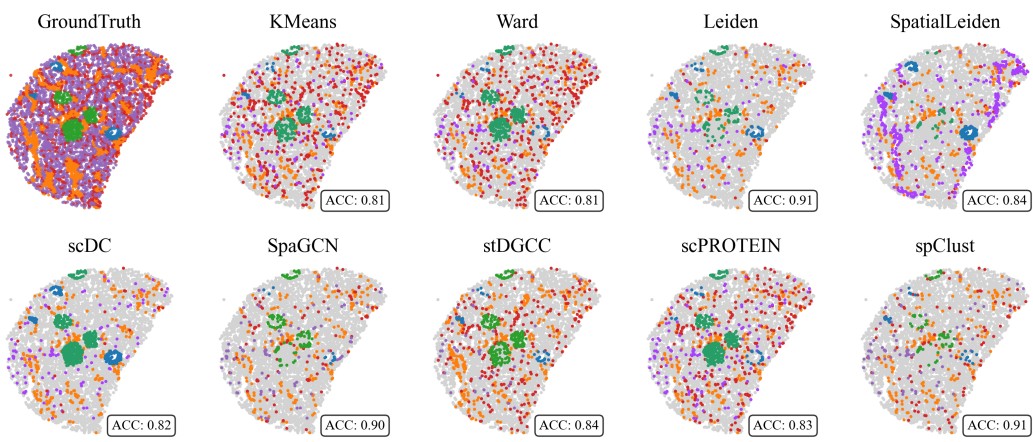

Figure D1: Visualization of spatial clustering results. Gray dots denote cases where clustering labels match the ground-truth labels, while dots of other colors denote the corresponding ground-truth labels when clustering labels do not match the ground-truths.

The differences between the spatial clustering results of different methods and the ground-truth results on the DKD_000 dataset are shown in Figure D1: gray dots represent correctly clustered cells,

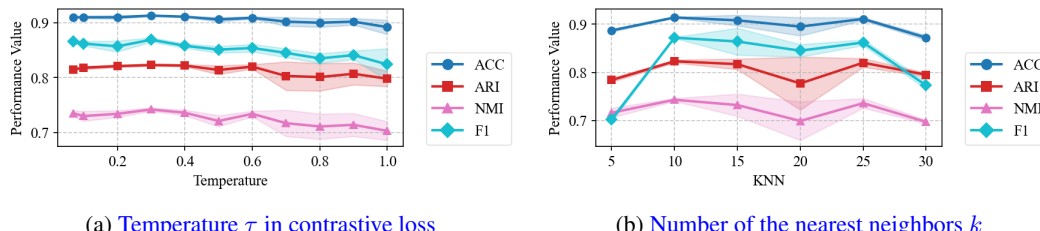

(a) Temperature $\tau$ in contrastive loss     (b) Number of the nearest neighbors $k$

Figure D2: Clustering performance of `spClust` with different hyperparameters on the DKD_000.

while dots of other colors represent incorrectly clustered ones. Among all methods, `spClust`, Leiden, and SpaGCN obviously achieve better spatial clustering performance. Except for scDC and SpatialLeiden, most methods can effectively cluster the major cell types. However, six methods (KMeans, Ward, SpatialLeiden, scDC, stDGCC, and scPROTEIN) all fail to accurately identify the minor cell types that are significantly clustered in space. Leiden demonstrates highly competitive results, owing to its ability to partition protein expression similarity graph and spatial microenvironment graph via modularity maximization, which fully reflects the potential of modularity maximization for clustering optimization. `spClust` achieves excellent clustering performance across all cell types, primarily because it addresses class imbalance and adopts a contrastive representation learning strategy with expression-spatial decoupling.

## D.2 HYPERPARAMETER ANALYSIS

To investigate the impact of different hyperparameters on the clustering performance of the `spClust`, we conducted hyperparameter sensitivity experiments. Specifically, the temperature hyperparameter $\tau$ of the contrastive loss was set to the range {0.07, 0.1, 0.2, 0.3, 0.4, 0.5, 0.6, 0.7, 0.8, 0.9, 1.0}, and the number of neighbors $k$ for constructing the cell $k$-NN graph was chosen from {5, 10, 15, 20, 25, 30}. As shown in Fig. a, on the DKD_000 dataset, the temperature hyperparameter $\tau$ has a limited effect on clustering performance. However, as the value of $\tau$ increases, the model's stability decreases. Regarding the neighbor number $k$, a small $k$ leads to insufficient modeling of local neighborhood relationships, while a large $k$ tends to aggravate feature over-smoothing.

## D.3 ADDITIONAL ROBUSTNESS ANALYSIS

Table D1: Clustering performance of `spClust` with different Initialized Pseudo-Labels on the Breast_6 and Melanoma_119 datasets.

| Dataset | Metric | Stage | Leiden with Different Seeds | | | Different Initialization Methods | | | Extreme Cases | | STDEV |
|---|---|---|---|---|---|---|---|---|---|---|---|
| | | | 0 | 42 | 2026 | SpatialLeiden | KMeans | Ward | Random | Ground-Truth | |
| Breast_6 | ACC | Initial | 0.765 | 0.765 | 0.765 | 0.589 | 0.773 | 0.776 | 0.342 | **1.000** | 0.189 |
| | | spClust | **0.816** | **0.806** | **0.859** | **0.840** | **0.874** | **0.805** | **0.794** | 0.861 | **0.030** |
| | ARI | Initial | 0.636 | **0.636** | 0.636 | 0.347 | 0.667 | **0.642** | -0.002 | **1.000** | 0.290 |
| | | spClust | **0.648** | 0.619 | **0.656** | **0.650** | **0.713** | 0.628 | **0.471** | 0.691 | **0.073** |
| | NMI | Initial | 0.484 | **0.484** | 0.484 | 0.413 | 0.522 | **0.488** | 0.001 | **1.000** | 0.269 |
| | | spClust | 0.480 | 0.430 | **0.491** | **0.509** | **0.533** | 0.446 | **0.416** | 0.538 | **0.046** |
| | F1 | Initial | 0.618 | 0.618 | 0.618 | 0.485 | 0.626 | 0.641 | 0.295 | **1.000** | 0.196 |
| | | spClust | **0.696** | **0.653** | **0.773** | **0.577** | **0.764** | **0.639** | **0.54** | 0.775 | **0.091** |
| Melanoma_119 | ACC | Initial | 0.426 | 0.424 | 0.422 | 0.527 | **0.530** | 0.479 | 0.123 | **1.000** | 0.242 |
| | | spClust | **0.550** | **0.613** | **0.501** | **0.615** | 0.527 | **0.512** | **0.501** | 0.563 | **0.046** |
| | ARI | Initial | 0.181 | 0.180 | 0.176 | 0.271 | 0.236 | 0.220 | 0.000 | **1.000** | 0.301 |
| | | spClust | **0.277** | **0.334** | **0.278** | **0.336** | **0.279** | **0.275** | **0.279** | 0.305 | **0.026** |
| | NMI | Initial | 0.253 | 0.252 | 0.246 | **0.316** | 0.262 | 0.264 | 0.003 | **1.000** | 0.289 |
| | | spClust | **0.315** | **0.319** | **0.337** | 0.287 | **0.295** | **0.302** | **0.312** | 0.328 | **0.017** |
| | F1 | Initial | 0.296 | 0.295 | **0.296** | 0.304 | 0.267 | 0.267 | 0.094 | **1.000** | 0.271 |
| | | spClust | **0.322** | **0.317** | 0.272 | **0.322** | **0.309** | **0.276** | **0.313** | 0.394 | **0.037** |

Table D1 further demonstrates the impact of varying-quality pseudo-labels on the clustering results of `spClust`. Compared with their performance on the DKD_000 dataset (as shown in Table 2), classical clustering methods yield poorer results on the Breast_6 and Melanoma_119 datasets, especially on Melanoma_119, suggesting that the initial pseudo-labels used in this experiment are of low quality. However, on these two datasets, `spClust` still achieves better outcomes than the initial clustering. Notably, when the pseudo-labels are completely random, `spClust` is only slightly affected, confirming its robustness.

# E   THE USE OF LARGE LANGUAGE MODELS

We promise that we did not use large language models (LLMs) in this work.

