# OpenReview forum: "Single-Cell Spatial Proteomics Clustering by Decoupling Spatiality and Expression"
_ICLR.cc/2026/Conference — Submitted to ICLR 2026_

### Official Review · Reviewer_73Tt · 2025-10-31

**Soundness:** 3
**Presentation:** 3
**Contribution:** 2
**Rating:** 4
**Confidence:** 2

**Summary:**

This paper proposes the spClust framework for clustering spatial proteomics data. The method combines spatial constrained oversampling (Spa-CSMOTE), dual-view contrastive learning, and module maximization clustering optimization to solve the class imbalance and space-expression view conflicts.

**Strengths:**

-Spa-CSMOTE incorporates spatial context into oversampling, avoiding synthesizing fake data.

-Based on scPROTEIN, a dual-view contrastive learning approach is proposed to address the heterogeneity of protein expression with the external environment.

-The technical details are relatively clear, and the code is open source.

**Weaknesses:**

-Only the kidney dataset was used, and the generalization to other tissues (such as tumors or brain tissue) was not validated.

-The Spa-CSMOTE relies on pseudo-labels generated by Leiden clustering, which may amplify errors if the initial clustering is inaccurate.

-Robustness strategies are also not discussed in the paper.

-Training time or resource consumption is not discussed, especially on larger-scale datasets.

-Hyperparameter sensitivity analysis is missing.

**Questions:**

Q1: How effective is Spa-CSMOTE in extremely imbalanced scenarios (e.g., minority class percentage <1%)? Is there a threshold to guide its application?

Q2: How do the selection of the temperature parameter $\tau$ and the number of neighbors K affect model performance?

Q3: Can the spClust handle larger-scale data or data from other scenarios?

Q4: Does the accuracy of pseudo-label generation significantly affect the clustering results of the spClust? If so, how should this be addressed?

---

> ### Author Response · Authors · 2025-11-22
> **Response to Reviewer 73Tt (Part 1/2)**
>
> Thanks for your valuable comments! Our responses are as follows:
>
> > ### **Effectiveness of Spa-CSMOTE in extremely imbalanced scenarios (Q1)**
>
> **A1:** Our experiments **already cover multiple extremely imbalanced datasets**, for example:
>
> - DKD\_004 dataset: The smallest class contains 57 cells, accounting for only 0.57% of the total number of cells;
> - DKD\_013 dataset: The smallest class contains 94 cells, accounting for 1.07% of the total number of cells.
>
> These datasets have been used to verify the performance of spClust, and the experimental results (Table 1 in the manuscript) can fully demonstrate the algorithm's effectiveness in extremely imbalanced scenarios.
>
> > ### **Hyperparameter analysis (Q2\&W5)**
>
> **A2:** We have supplemented sensitivity experiments on the temperature hyperparameter $\tau$ and the number of neighbors K. The K value dictates the node neighbor count in the KNN graph: an overly large K induces over-smoothing, while an overly small K causes insufficient neighbor modeling. Conventionally, K is recommended to be set to ~15, with adjustments tailored to data characteristics. The temperature hyperparameter is parameterized as a learnable weight with an initial value.
>
> **Table R1**: Clustering performance with different K on the DKD\_kidney\_000.
>
> |Metric|5|10|15|20|25|30|
> |-|-|-|-|-|-|-|
> |**ACC**|0.887|0.914|0.908|0.895|0.911|0.872|
> |**ARI**|0.784|0.823|0.817|0.777|0.820|0.795|
> |**NMI**|0.716|0.743|0.732|0.699|0.736|0.697|
> |**F1**|0.702|0.872|0.864|0.845|0.862|0.773|
>
> **Table R2**: Clustering performance with different $\tau$ on the DKD\_kidney\_000.
>
> |$\tau$|0.07|0.1|0.2|0.3|0.4|0.5|0.6|0.7|0.8|0.9|1.0|
> |-|-|-|-|-|-|-|-|-|-|--|-|
> |**ACC**|0.910|0.910|0.910|0.913|0.911|0.906|0.909|0.902|0.900|0.902|0.892|
> |**ARI**|0.815|0.818|0.821|0.823|0.822|0.814|0.820|0.803|0.801|0.807|0.798|
> |**NMI**|0.735|0.730|0.734|0.742|0.736|0.721|0.734|0.717|0.711|0.714|0.703|
> |**F1**|0.866|0.862|0.857|0.869|0.858|0.851|0.854|0.845|0.835|0.841|0.824|
>
> > ### **Complexity analysis and other scenarios (Q3\&W1\&W4)**
>
> **A3:** The spClust consists of the following key steps:
>
> Leiden clustering initialization → Spa-CSMOTE oversampling → KNN graph construction → GAT-based feature extraction → loss computation. The time complexity of each step is summarized in the table below:
>
> **Table R3**: Computational complexity of each module in spClust.
>
> |Leiden|Spa-CSMOTE|KNN Graph|GAT|Contrastive|Modularity|
> |--|--|--|--|--|--|
> |$\mathcal{O}(n)$|$\mathcal{O}(\sum_{i=1}^km_i(n+m_id))$|$\mathcal{O}(n^2d)$|$\mathcal{O}(n\sum_{i=1}^3d_{i-1}di+\|\mathcal{E}\|\sum_{i=1}^{3}d_i)$|$\mathcal{O}(n^2)$|$\mathcal{O}(n^2)$|
>
> Additionally, we have supplemented the time and space costs. Due to the need to generate a large number of cells for imbalanced data, our method incurs certain memory and time costs. Compared with stDGCC, the performance gain offset by the resource consumption is acceptable.
>
> **Table R4**: Time and space costs of different methods on the DKD\_kidney\_000.
>
> |Metric|KMeans|Ward|Leiden|SpatialLeiden|scDC|SpaGCN|stDGCC|scPROTEIN|Ours|
> |-|-|-|-|-|-|-|-|-|-|
> |**Time(s)**|0.14|0.55|1.59|9.25|18.12|6.86|84.28|8.26|93.07|
> |**Peak CPU(MB)/GPU(MB)**|303/NA|273/NA|392/NA|919/NA|1608/43|1441/NA|1743/219|1783/843|9882/4869|
>
> Regarding the algorithm's applicability in other scenarios, **we have supplemented breast cancer [1] and melanoma datasets [2]**, each containing 3 tissue sections. These datasets also exhibit significant class imbalance and face the challenge of heterogeneity between spatial proximity and expression similarity. Experimental results demonstrate that our method still achieves remarkable performance improvements in such scenarios.
>
> [1] Esther Danenberg, et al. Breast tumor microenvironment structures are associated with genomic features and clinical outcome. Nat. Genet., 54(5):660–669, 2022
>
> [2] Tobias Hoch, et al. Multiplexed imaging mass cytometry of the chemokine milieus in melanoma characterizes features of the response to immunotherapy. Sci. Immunol., 7(70):eabk1692, 2022.

---

> ### Author Response · Authors · 2025-11-22
> **Response to Reviewer 73Tt (Part 2/2)**
>
> > ### **Robustness of pseudo-labels (Q4\&W2\&W3)**
>
> **A4:** The innovations of spClust and our contributions are mainly reflected in two aspects:
>
> 1) We focus on the overlooked yet **critical and challenging class imbalance issue** in single-cell spatial proteomics, and propose the spClust to mitigate its interference on feature learning and clustering performance.
>
> 2) We also focus on **the heterogeneity between protein expression and spatial location**, and propose a decoupled strategy for extracting expression features and spatial features, which effectively reduces the interference of cell heterogeneity on clustering.
>
> > ### **Robustness to noise in the pseudo-labels (W2)**
>
> **A5:**  **spClust is robust and pseudo-labels have only a slight impact**. The initialized pseudo-labels are related to the imbalance judgment and Spa-CSMOTE oversampling directly. However, *although the so-called pseudo-labels are "fake" in form, the data distribution they represent is real*; **our method is data-driven rather than label-dependent**, and we *enhance the robustness of spClust through periodic updates of pseudo-labels and prototypes*. We verified the robustness of spClust through extensive experiments:
>
> 1) Pseudo-labels initialized by different clustering methods on the DKD\_kidney\_000: Results show that spClust outperforms Leiden/SpatialLeiden/KMeans/Ward; though it exhibits certain fluctuations under different initialization clustering methods, **its robustness is superior to other methods**.
>
> **Table R5**: Comparison of different pseudo-label initialization methods on the DKD\_kidney\_000.
>
> |Method|Metric|Leiden|SpatialLeiden|KMeans|Ward| STDEV|
> |-|-|-|-|-|-|-|
> | **Initial**| **ACC** | 0.905| 0.715| 0.742| 0.814| 0.085|
> || **ARI** | 0.801| 0.483| 0.654| 0.762| 0.142|
> || **NMI** | 0.744| 0.563| 0.582| 0.624| 0.081|
> || **F1**  | 0.855| 0.718| 0.565| 0.640| 0.124|
> | **spClust (Ours)** | **ACC** | **0.912** | **0.905**| **0.895** | **0.849** | **0.028** (↑67%) |
> || **ARI** | **0.821** | **0.819**| **0.807** | **0.789** | **0.015** (↑89%) |
> || **NMI** | **0.746** | **0.729**| **0.716** | **0.690** | **0.024** (↑70%) |
> || **F1**  | **0.866** | **0.838**| **0.812** | **0.747** | **0.051** (↑59%) |
>
> 2) Pseudo-labels initialized by Leiden clustering with different random seeds: **Results demonstrate that Leiden clustering exhibits excellent robustness, and our method initialized with it maintains this property**.
>
> **Table R6**: Comparison of pseudo-labels initialized by Leiden with different random seeds on the DKD\_kidney\_000.
>
> | Method| Metric| seed=0 | seed=42 | seed=2026 |
> |-|-|-|-|-|
> | **Initial**| **ACC**| 0.906|0.905| 0.907|
> || **ARI** | 0.804 | 0.801| 0.806|
> || **NMI** | 0.746  | 0.744| 0.748|
> || **F1**  | 0.856  | 0.855| 0.856|
> | **spClust (Ours)** | **ACC** | 0.912| 0.912| 0.913|
> || **ARI** | 0.820  | 0.821| 0.822|
> || **NMI** | 0.744  | 0.746| 0.746|
> || **F1**  | 0.868  | 0.866| 0.866|
>
> 3) Pseudo-labels under extreme cases (i.e., random labels or ground-truth): Results show that under random labels (completely mismatched with the data similarity distribution), our method still enables effective clustering. Attribute this capability to GAT-based feature extraction and periodic updates of prototypes and pseudo-labels. Notably, even with ground-truth labels (perfectly matching the data), it does not yield optimal clustering performance, affirming that our method is data-driven, not label-driven. The so-called "pseudo-labels" actually encode the real data distribution.
>
> **Table R7**: Comparison of clustering results under extreme pseudo-labeling cases on the DKD\_kidney\_000.
>
> | Method| Metric| Random | Ground-Truth |
> | -| - |-|- |
> | **Initial**| **ACC** | 0.209  | 1.000|
> || **ARI** | 0.000  | 1.000|
> || **NMI** | 0.001  | 1.000|
> || **F1**  | 0.177  | 1.000|
> | **spClust (Ours)** | **ACC** | 0.687  | 0.907|
> || **ARI** | 0.499  | 0.821|
> || **NMI** | 0.600  | 0.735|
> || **F1**  | 0.477  | 0.849|
>
> 4) Pseudo-labels and prototypes correction ability of spClust: The robustness of our method depends primarily on periodic prototype and pseudo-label updates. Removing both leads to a significant performance drop. Notably, the removal of prototype updates has the most pronounced impact, while removing pseudo-label updates causes no significant performance degradation.
>
> **Table R8**: Ablation study on the periodic prototypes and pseudo-labels update strategy on the DKD\_kidney\_000.
>
> | Metric| spClust| w/o pseudo-labels update | w/o prototypes update | w/o both|
> | - |-| -| - | -|
> | **ACC** | 0.914 | 0.908| 0.770| 0.779 |
> | **ARI** | 0.823| 0.818| 0.557| 0.573 |
> | **NMI** | 0.740| 0.728| 0.549| 0.570 |
> | **F1**  | 0.870| 0.853| 0.695| 0.717 |
>
> Thank you again for your valuable comments! All the additional results will be incorporated into the manuscript.

---

### Official Review · Reviewer_Bag2 · 2025-10-31

**Soundness:** 3
**Presentation:** 3
**Contribution:** 2
**Rating:** 4
**Confidence:** 5

**Summary:**

The paper proposes a spatial transcriptomics data clustering method that addresses the issue of cluster imbalance through an oversampling strategy. In addition, a cross-view contrastive learning network is introduced to learn robust embeddings from augmented data for improved clustering performance. Extensive experiments on multiple datasets are conducted to evaluate the effectiveness of the proposed approach.

**Strengths:**

1. A spatial transcriptomics data clustering method with an integrated oversampling strategy is proposed to address the problem of cluster imbalance.

2. A cross-view contrastive learning network is developed to learn robust embeddings from augmented data for improved clustering performance.

3. Comprehensive experiments on multiple datasets are conducted to validate the effectiveness of the proposed approach.

**Weaknesses:**

1. The novelty of the proposed method appears limited. Several techniques adopted in this work, such as oversampling, cross-view contrastive learning, and label modularity maximisation loss, have been widely used in previous representation learning and clustering studies.

2. Regarding the pseudo-label-guided data augmentation, if the pseudo-labels are of poor quality, how can the proposed approach ensure that the augmentation process remains effective?

3. The differences between Equations (3) and (4) for spatial position augmentation are unclear. Additionally, the rationale behind the superior performance of the latter formulation should be further explained.

4. The computational complexity of the proposed method is not analysed or discussed.

**Questions:**

See Weaknesses

---

> ### Author Response · Authors · 2025-11-22
> **Response to Reviewer Bag2 (Part 1/2)**
>
> Thanks for your valuable comments! Our responses are as follows:
>
> > ### **Novelty clarification (W1)**
>
> **A1:** The innovations of spClust and our contributions are mainly reflected in two aspects:
>
> 1) We focus on the overlooked yet **critical and challenging class imbalance issue** in single-cell spatial proteomics, and propose the spClust to mitigate its interference on feature learning and clustering performance.
>
> 2) We also focus on **the heterogeneity between protein expression and spatial location**, and propose a decoupled strategy for extracting expression features and spatial features, which effectively reduces the interference of cell heterogeneity on clustering.
>
> > ### **Robustness to noise in the pseudo-labels (W2)**
>
> **A2:**  **spClust is robust and pseudo-labels have only a slight impact**. The initialized pseudo-labels are related to the imbalance judgment and Spa-CSMOTE oversampling directly. However, *although the so-called pseudo-labels are "fake" in form, the data distribution they represent is real*; **our method is data-driven rather than label-dependent**, and we *enhance the robustness of spClust through periodic updates of pseudo-labels and prototypes*. We verified the robustness of spClust through extensive experiments:
>
> 1) Pseudo-labels initialized by different clustering methods on the DKD\_kidney\_000: Results show that spClust outperforms Leiden/SpatialLeiden/KMeans/Ward; though it exhibits certain fluctuations under different initialization clustering methods, **its robustness is superior to other methods**.
>
> **Table R1**: Comparison of different pseudo-label initialization methods on the DKD\_kidney\_000.
>
> |Method|Metric|Leiden|SpatialLeiden|KMeans|Ward| STDEV|
> |-|-|-|-|-|-|-|
> | **Initial**| **ACC** | 0.905| 0.715| 0.742| 0.814| 0.085|
> || **ARI** | 0.801| 0.483| 0.654| 0.762| 0.142|
> || **NMI** | 0.744| 0.563| 0.582| 0.624| 0.081|
> || **F1**  | 0.855| 0.718| 0.565| 0.640| 0.124|
> | **spClust (Ours)** | **ACC** | **0.912** | **0.905**| **0.895** | **0.849** | **0.028** (↑67%) |
> || **ARI** | **0.821** | **0.819**| **0.807** | **0.789** | **0.015** (↑89%) |
> || **NMI** | **0.746** | **0.729**| **0.716** | **0.690** | **0.024** (↑70%) |
> || **F1**  | **0.866** | **0.838**| **0.812** | **0.747** | **0.051** (↑59%) |
>
> 2) Pseudo-labels initialized by Leiden clustering with different random seeds: **Results demonstrate that Leiden clustering exhibits excellent robustness, and our method initialized with it maintains this property**.
>
> **Table R2**: Comparison of pseudo-labels initialized by Leiden with different random seeds on the DKD\_kidney\_000.
>
> | Method| Metric| seed=0 | seed=42 | seed=2026 |
> |-|-|-|-|-|
> | **Initial**| **ACC**| 0.906|0.905| 0.907|
> || **ARI** | 0.804 | 0.801| 0.806|
> || **NMI** | 0.746  | 0.744| 0.748|
> || **F1**  | 0.856  | 0.855| 0.856|
> | **spClust (Ours)** | **ACC** | 0.912| 0.912| 0.913|
> || **ARI** | 0.820  | 0.821| 0.822|
> || **NMI** | 0.744  | 0.746| 0.746|
> || **F1**  | 0.868  | 0.866| 0.866|
>
> 3) Pseudo-labels under extreme cases (i.e., random labels or ground-truth): Results show that under random labels (completely mismatched with the data similarity distribution), our method still enables effective clustering. Attribute this capability to GAT-based feature extraction and periodic updates of prototypes and pseudo-labels. Notably, even with ground-truth labels (perfectly matching the data), it does not yield optimal clustering performance, affirming that our method is data-driven, not label-driven. The so-called "pseudo-labels" actually encode the real data distribution.
>
> **Table R3**: Comparison of clustering results under extreme pseudo-labeling cases on the DKD\_kidney\_000.
>
> | Method| Metric| Random | Ground-Truth |
> | -| - |-|- |
> | **Initial**| **ACC** | 0.209  | 1.000|
> || **ARI** | 0.000  | 1.000|
> || **NMI** | 0.001  | 1.000|
> || **F1**  | 0.177  | 1.000|
> | **spClust (Ours)** | **ACC** | 0.687  | 0.907|
> || **ARI** | 0.499  | 0.821|
> || **NMI** | 0.600  | 0.735|
> || **F1**  | 0.477  | 0.849|
>
> 4) Pseudo-labels and prototypes correction ability of spClust: The robustness of our method depends primarily on periodic prototype and pseudo-label updates. Removing both leads to a significant performance drop. Notably, the removal of prototype updates has the most pronounced impact, while removing pseudo-label updates causes no significant performance degradation.
>
> **Table R4**: Ablation study on the periodic prototypes and pseudo-labels update strategy on the DKD\_kidney\_000.
>
> | Metric| spClust| w/o pseudo-labels update | w/o prototypes update | w/o both|
> | - |-| -| - | -|
> | **ACC** | 0.914 | 0.908| 0.770| 0.779 |
> | **ARI** | 0.823| 0.818| 0.557| 0.573 |
> | **NMI** | 0.740| 0.728| 0.549| 0.570 |
> | **F1**  | 0.870| 0.853| 0.695| 0.717 |

---

> ### Author Response · Authors · 2025-11-22
> **Response to Reviewer Bag2 (Part 2/2)**
>
> > ### **Differences between Equations (3) and (4) (W3)**
>
> **A3:** The main difference between Eq. (3) and Eq. (4) is that the interpolation weights $\lambda_ 1$ in Eq. (3) and $\lambda _2$ in Eq. (4) are constrained with different conditions. Specifically, $\lambda _1$ in Eq. (3) follows a uniform distribution $\lambda _1 \sim Uniform(0,1)$, resulting in a random spatial distribution of synthetic cells. However, **a better spatial location of synthetic cells should be close to the two original cells** used to generate the synthetic cell. Therefore, $\lambda _2$ in Eq. (4) has a higher probability to close to 0 or 1. We respectfully remind that **the detailed explanation of this design has been clearly described in the manuscript (after Eq. 4)**. To more intuitively demonstrate the effectiveness of this design, we have added to the appendix a visualization of the spatial location of synthetic cells under different conditions. Here we show the comparison of the clustering results of Eq. (3) and Eq. (4) on the DKD\_kidney\_000.
>
> **Table R5**: Comparison of different spatial constraints on the DKD\_kidney\_000.
>
> ||Eq. (4)|Eq. (3)|
> |-|-|-|
> |**ARI**|0.822|0.819|
>
> > ### **Complexity and analysis (W4)**
>
> **A4:** The spClust consists of the following key steps:
>
> Leiden clustering initialization → Spa-CSMOTE oversampling → KNN graph construction → GAT-based feature extraction → loss computation. The time complexity of each step is summarized in the table below:
>
> **Table R6**: Computational complexity of each module in spClust.
>
> |Leiden|Spa-CSMOTE|KNN Graph|GAT|Contrastive|Modularity|
> |--|--|--|--|--|--|
> |$\mathcal{O}(n)$|$\mathcal{O}(\sum_{i=1}^km_i(n+m_id))$|$\mathcal{O}(n^2d)$|$\mathcal{O}(n\sum_{i=1}^3d_{i-1}di+\|\mathcal{E}\|\sum_{i=1}^{3}d_i)$|$\mathcal{O}(n^2)$|$\mathcal{O}(n^2)$|
>
> Additionally, we have supplemented the time and space costs. Due to the need to generate a large number of cells for imbalanced data, our method incurs certain memory and time costs. Compared with stDGCC, the performance gain offset by the resource consumption is acceptable.
>
> **Table R7**: Time and space costs of different methods on the DKD\_kidney\_000.
>
> |Metric|KMeans|Ward|Leiden|SpatialLeiden|scDC|SpaGCN|stDGCC|scPROTEIN|Ours|
> |-|-|-|-|-|-|-|-|-|-|
> |**Time(s)**|0.14|0.55|1.59|9.25|18.12|6.86|84.28|8.26|93.07|
> |**Peak CPU(MB)/GPU(MB)**|303/NA|273/NA|392/NA|919/NA|1608/43|1441/NA|1743/219|1783/843|9882/4869|
>
> Thank you again for your valuable comments! All the additional results will be incorporated into the manuscript.

---

> > ### Comment · Reviewer_Bag2 · 2025-11-28
> > **Response to Authors**
> >
> > Thanks to the authors for their reply. The experimental results show that the proposed method is generally insensitive to the pseudo-labelling strategy on datasets that are relatively easy to cluster (e.g., F1 > 85%). However, for more challenging datasets, it remains unclear whether the method consistently performs well. In addition, I suggest that the authors provide further explanation on how the proposed approach maintains effective augmentations when the pseudo-labels are of low quality.

---

> > > ### Author Response · Authors · 2025-12-01
> > >
> > > Thank you for your insightful comments. We have incorporated two challenging datasets (Breast\_6 and Melanoma\_119) to further validate the robustness of spClust, with the results of extensive experiments presented in Tables R8-R13. All findings confirm that spClust maintains strong robustness against the quality of initial pseudo-labels. We hereby re-clarify the reasons for this robustness:
> > >
> > > 1. Pseudo-labels essentially serve as an unsupervised grouping tool, reflecting the distribution characteristics of the grouped data. Even if pseudo-labels are inaccurate, as long as they improve coverage of the data distribution, they will exert a positive impact on subsequent feature learning. For completely random label grouping, the generated subgroups are relatively balanced in size, eliminating the need for excessive oversampling and thus avoiding data distribution distortion. Clustering-based pseudo-label grouping is performed based on the intrinsic similarity between data samples, which also preserves the inherent data distribution.
> > >
> > > 2. The initial pseudo-labels are no longer used in subsequent processes. Instead, during the feature learning phase, pseudo-labels and clustering prototypes are periodically updated based on the latest clustering results, preventing the propagation of errors from initial pseudo-labels.
> > >
> > > We hope the above explanations effectively address your concerns.
> > >
> > > **Table R8**: Comparison of different pseudo-label initialization methods on the Breast\_6.
> > >
> > > |Method|Metric|Leiden|SpatialLeiden|KMeans|Ward|STDEV|
> > > |-|-|-|-|-|-|-|
> > > |**Initial**|**ACC**|0.765|0.589|0.773|0.776|0.091|
> > > ||**ARI**|**0.636**|0.347|0.667|**0.642**|0.151|
> > > ||**NMI**|**0.484**|0.413|0.522|**0.488**|**0.046**|
> > > ||**F1**|0.618|0.485|0.626|0.641|**0.072**|
> > > |**spClust (Ours)**|**ACC**|**0.806**|**0.840**|**0.874**|**0.805**|**0.033** (↑64\%)|
> > > ||**ARI**|0.619|**0.650**|**0.713**|0.628|**0.042** (↑ 72\%)|
> > > ||**NMI**|0.430|**0.509**|**0.533**|0.446|0.049 (↓7%)|
> > > ||**F1**|**0.653**|**0.577**|**0.764**|**0.639**|0.078 (↓8\%)|
> > >
> > > **Table R9**: Comparison of different pseudo-label initialization methods on the Melanoma\_119.
> > >
> > > |Method|Metric|Leiden|SpatialLeiden|KMeans|Ward|STDEV|
> > > |-|-|-|-|-|-|-|
> > > |**Initial**|**ACC**|0.424|0.527|**0.530**|0.479|**0.050**|
> > > ||**ARI**|0.180|0.271|0.236|0.220|0.038|
> > > ||**NMI**|0.252|**0.316**|0.262|0.264|0.029|
> > > ||**F1**|0.295|0.304|0.267|0.267|**0.019**|
> > > |**spClust (Ours)**|**ACC**|**0.613**|**0.615**|0.527|**0.512**|0.055 (↓10\%)|
> > > ||**ARI**|**0.334**|**0.336**|**0.279**|**0.275**|**0.034** (↑11\%)|
> > > ||**NMI**|**0.319**|0.287|**0.295**|**0.302**|**0.014** (↑52\%)|
> > > ||**F1**|**0.317**|**0.322**|**0.309**|**0.276**|0.021 (↓10\%)|
> > >
> > > **Table R10**: Comparison of pseudo-labels initialized by Leiden with different random seeds on the Breast\_6.
> > >
> > > |Method|Metric|seed=0|seed=42|seed=2026|
> > > |-|-|-|-|-|
> > > |**Initial**|**ACC**|0.765|0.765|0.765|
> > > ||**ARI**|0.636|0.636|0.636|
> > > ||**NMI**|0.484|0.484|0.484|
> > > ||**F1**|0.618|0.618|0.618|
> > > |**spClust (Ours)**|**ACC**|0.816|0.806|0.859|
> > > ||**ARI**|0.648|0.619|0.656|
> > > ||**NMI**|0.480|0.430|0.491|
> > > ||**F1**|0.696|0.653|0.773|
> > >
> > > **Table R11**: Comparison of pseudo-labels initialized by Leiden with different random seeds on the Melanoma\_119.
> > >
> > > |Method|Metric|seed=0|seed=42|seed=2026|
> > > |-|-|-|-|-|
> > > |**Initial**|**ACC**|0.426|0.424|0.422|
> > > ||**ARI**|0.181|0.180|0.176|
> > > ||**NMI**|0.253|0.252|0.246|
> > > ||**F1**|0.296|0.295|0.296|
> > > |**spClust (Ours)**|**ACC**|0.550|0.613|0.501|
> > > ||**ARI**|0.277|0.334|0.278|
> > > ||**NMI**|0.315|0.319|0.337|
> > > ||**F1**|0.322|0.317|0.272|
> > >
> > > **Table R12**: Comparison of clustering results under extreme pseudo-labeling cases on the Breast\_6.
> > >
> > > |Method|Metric|Random|Ground-Truth|
> > > |-|-|-|-|
> > > |**Initial**|**ACC**|0.342|1.000|
> > > ||**ARI**|−0.002|1.000|
> > > ||**NMI**|0.001|1.000|
> > > ||**F1**|0.295|1.000|
> > > |**spClust (Ours)**|**ACC**|0.794|0.861|
> > > ||**ARI**|0.471|0.691|
> > > ||**NMI**|0.416|0.538|
> > > ||**F1**|0.540|0.775|
> > >
> > > **Table R13**: Comparison of clustering results under extreme pseudo-labeling cases on the Melanoma\_119.
> > >
> > > |Method|Metric|Random|Ground-Truth|
> > > |-|-|-|-|
> > > |**Initial**|**ACC**|0.123|1.000|
> > > ||**ARI**|0.000|1.000|
> > > ||**NMI**|0.003|1.000|
> > > ||**F1**|0.094|1.000|
> > > |**spClust (Ours)**|**ACC**|0.501|0.563|
> > > ||**ARI**|0.279|0.305|
> > > ||**NMI**|0.312|0.328|
> > > ||**F1**|0.313|0.394|

---

### Official Review · Reviewer_3aV3 · 2025-11-01

**Soundness:** 2
**Presentation:** 3
**Contribution:** 3
**Rating:** 4
**Confidence:** 3

**Summary:**

This paper proposes spClust, an unsupervised clustering framework for single-cell spatial proteomics that decouples spatial and expression information. spClust consists of three stages. It first introduces Spa-CSMOTE, a spatially constrained oversampling method that uses pseudo-labels generated by the Leiden algorithm to alleviate class imbalance. Next, a cross-view contrastive learning module jointly aligns spatial and expression embeddings to ensure consistent representations across the two different modalities. Finally, soft-label modularity maximization is applied to optimize cluster assignments in an end-to-end manner through a differentiable relaxation of graph modularity. Experiments demonstrate that spClust achieves more accurate clustering results than existing methods.

**Strengths:**

1. The proposed method combines spatially constrained oversampling, cross-view contrastive learning, and soft-label modularity optimization into a unified end-to-end system.

2. Figure 1 is particularly well designed and effectively illustrates the overall framework.

3. The paper is clearly written and well-organized

**Weaknesses:**

1. The paper lacks analysis on computational complexity and scalability of the proposed method.
2. The soft-label modularity maximization objective is a straightforward adaptation of prior differentiable modularity-based clustering losses. It seems that this paper has limited theoretical novelty.
3. Since the proposed method relies on pseudo-labels obtained by Leiden Clustering, the method’s stability and performance could be sensitive to the initial clustering quality. The paper lacks an ablation or robustness analysis for this dependency. In cases where the initial pseudo-labels are inaccurate, the performance of spClust may degrade substantially.

**Questions:**

Q1. Could the authors provide empirical evidence or sensitivity analysis demonstrating the robustness of spClust to inaccurate pseudo-labels or low-quality initial clusterings?

Q2. Could the authors clearly articulate the technical novelty or theoretical contribution beyond combining the multiple ideas?

---

> ### Author Response · Authors · 2025-11-22
> **Response to Reviewer 3aV3 (Part 1/2)**
>
> Thanks for your valuable comments! Our responses are as follows:
>
> > ### **Robustness to noise in the pseudo-labels (Q1\&W3)**
>
> **A1:**  **spClust is robust and pseudo-labels have only a slight impact**. The initialized pseudo-labels are related to the imbalance judgment and Spa-CSMOTE oversampling directly. However, *although the so-called pseudo-labels are "fake" in form, the data distribution they represent is real*; **our method is data-driven rather than label-dependent**, and we *enhance the robustness of spClust through periodic updates of pseudo-labels and prototypes*. We verified the robustness of spClust through extensive experiments:
>
> 1) Pseudo-labels initialized by different clustering methods on the DKD\_kidney\_000: Results show that spClust outperforms Leiden/SpatialLeiden/KMeans/Ward; though it exhibits certain fluctuations under different initialization clustering methods, **its robustness is superior to other methods**.
>
> **Table R1**: Comparison of different pseudo-label initialization methods on the DKD\_kidney\_000.
>
> |Method|Metric|Leiden|SpatialLeiden|KMeans|Ward| STDEV|
> |-|-|-|-|-|-|-|
> | **Initial**| **ACC** | 0.905| 0.715| 0.742| 0.814| 0.085|
> || **ARI** | 0.801| 0.483| 0.654| 0.762| 0.142|
> || **NMI** | 0.744| 0.563| 0.582| 0.624| 0.081|
> || **F1**  | 0.855| 0.718| 0.565| 0.640| 0.124|
> | **spClust (Ours)** | **ACC** | **0.912** | **0.905**| **0.895** | **0.849** | **0.028** (↑67%) |
> || **ARI** | **0.821** | **0.819**| **0.807** | **0.789** | **0.015** (↑89%) |
> || **NMI** | **0.746** | **0.729**| **0.716** | **0.690** | **0.024** (↑70%) |
> || **F1**  | **0.866** | **0.838**| **0.812** | **0.747** | **0.051** (↑59%) |
>
> 2) Pseudo-labels initialized by Leiden clustering with different random seeds: **Results demonstrate that Leiden clustering exhibits excellent robustness, and our method initialized with it maintains this property**.
>
> **Table R2**: Comparison of pseudo-labels initialized by Leiden with different random seeds on the DKD\_kidney\_000.
>
> | Method| Metric| seed=0 | seed=42 | seed=2026 |
> |-|-|-|-|-|
> | **Initial**| **ACC**| 0.906|0.905| 0.907|
> || **ARI** | 0.804 | 0.801| 0.806|
> || **NMI** | 0.746  | 0.744| 0.748|
> || **F1**  | 0.856  | 0.855| 0.856|
> | **spClust (Ours)** | **ACC** | 0.912| 0.912| 0.913|
> || **ARI** | 0.820  | 0.821| 0.822|
> || **NMI** | 0.744  | 0.746| 0.746|
> || **F1**  | 0.868  | 0.866| 0.866|
>
> 3) Pseudo-labels under extreme cases (i.e., random labels or ground-truth): Results show that under random labels (completely mismatched with the data similarity distribution), our method still enables effective clustering. Attribute this capability to GAT-based feature extraction and periodic updates of prototypes and pseudo-labels. Notably, even with ground-truth labels (perfectly matching the data), it does not yield optimal clustering performance, affirming that our method is data-driven, not label-driven. The so-called "pseudo-labels" actually encode the real data distribution.
>
> **Table R3**: Comparison of clustering results under extreme pseudo-labeling cases on the DKD\_kidney\_000.
>
> | Method| Metric| Random | Ground-Truth |
> | -| - |-|- |
> | **Initial**| **ACC** | 0.209  | 1.000|
> || **ARI** | 0.000  | 1.000|
> || **NMI** | 0.001  | 1.000|
> || **F1**  | 0.177  | 1.000|
> | **spClust (Ours)** | **ACC** | 0.687  | 0.907|
> || **ARI** | 0.499  | 0.821|
> || **NMI** | 0.600  | 0.735|
> || **F1**  | 0.477  | 0.849|
>
> 4) Pseudo-labels and prototypes correction ability of spClust: The robustness of our method depends primarily on periodic prototype and pseudo-label updates. Removing both leads to a significant performance drop. Notably, the removal of prototype updates has the most pronounced impact, while removing pseudo-label updates causes no significant performance degradation.
>
> **Table R4**: Ablation study on the periodic prototypes and pseudo-labels update strategy on the DKD\_kidney\_000.
>
> | Metric| spClust| w/o pseudo-labels update | w/o prototypes update | w/o both|
> | - |-| -| - | -|
> | **ACC** | 0.914 | 0.908| 0.770| 0.779 |
> | **ARI** | 0.823| 0.818| 0.557| 0.573 |
> | **NMI** | 0.740| 0.728| 0.549| 0.570 |
> | **F1**  | 0.870| 0.853| 0.695| 0.717 |

---

> ### Author Response · Authors · 2025-11-22
> **Response to Reviewer 3aV3 (Part 2/2)**
>
> > ### **Novelty and contributions (Q2\&W2)**
>
> **A2:** The innovations of spClust and our contributions are mainly reflected in two aspects:
>
> 1) We focus on the overlooked yet **critical and challenging class imbalance issue** in single-cell spatial proteomics, and propose the spClust to mitigate its interference on feature learning and clustering performance.
>
> 2) We also focus on **the heterogeneity between protein expression and spatial location**, and propose a decoupled strategy for extracting expression features and spatial features, which effectively reduces the interference of cell heterogeneity on clustering.
>
> > ### **Complexity and scalability analysis (W1)**
>
> **A3:** The spClust consists of the following key steps:
>
> Leiden clustering initialization → Spa-CSMOTE oversampling → KNN graph construction → GAT-based feature extraction → loss computation. The time complexity of each step is summarized in the table below:
>
> **Table R5**: Computational complexity of each module in spClust.
>
> |Leiden|Spa-CSMOTE|KNN Graph|GAT|Contrastive|Modularity|
> |--|--|--|--|--|--|
> |$\mathcal{O}(n)$|$\mathcal{O}(\sum_{i=1}^km_i(n+m_id))$|$\mathcal{O}(n^2d)$|$\mathcal{O}(n\sum_{i=1}^3d_{i-1}di+\|\mathcal{E}\|\sum_{i=1}^{3}d_i)$|$\mathcal{O}(n^2)$|$\mathcal{O}(n^2)$|
>
> Additionally, we have supplemented the time and space costs. Due to the need to generate a large number of cells for imbalanced data, our method incurs certain memory and time costs. Compared with stDGCC, the performance gain offset by the resource consumption is acceptable.
>
> **Table R6**: Time and space costs of different methods on the DKD\_kidney\_000.
>
> |Metric|KMeans|Ward|Leiden|SpatialLeiden|scDC|SpaGCN|stDGCC|scPROTEIN|Ours|
> |-|-|-|-|-|-|-|-|-|-|
> |**Time(s)**|0.14|0.55|1.59|9.25|18.12|6.86|84.28|8.26|93.07|
> |**Peak CPU(MB)/GPU(MB)**|303/NA|273/NA|392/NA|919/NA|1608/43|1441/NA|1743/219|1783/843|9882/4869|
>
> Thank you again for your valuable comments! All the additional results will be incorporated into the manuscript.

---

### Official Review · Reviewer_iug6 · 2025-11-03

**Soundness:** 2
**Presentation:** 3
**Contribution:** 2
**Rating:** 4
**Confidence:** 4

**Summary:**

This paper proposes a deep clustering framework named spClust for single-cell spatial proteomics data analysis. The method mitigates cell-type imbalance through a spatially constrained oversampling technique (Spa-CSMOTE) and employs decoupled dual-view contrastive learning to integrate spatial adjacency and expression similarity information. Ultimately, it enhances the accuracy and biological plausibility of cell clustering through end-to-end clustering optimization.

**Strengths:**

The main strengths of the paper lie in its innovative integration of spatially constrained oversampling and multi-view contrastive learning, which are specifically reflected in: the proposal of the Spa-CSMOTE method to generate biologically plausible virtual minority cells in an unsupervised setting; the construction of a dual-view graph structure based on expression similarity and spatial adjacency, along with feature decoupling and alignment via a weight-sharing graph attention network; and the design of an end-to-end clustering objective that incorporates dynamic prototype learning and soft-label modularity maximization, effectively enhancing the quality of cell embeddings and clustering performance.

**Weaknesses:**

1. The method first uses Leiden to obtain pseudo-labels for identifying class imbalance, then performs oversampling based on these labels. If the initial pseudo-labels are biased, subsequent Spa-CSMOTE and contrastive learning could amplify this bias, leading to "error amplification." The paper lacks a systematic evaluation of robustness to noise in the pseudo-labels.
2. UMAP itself has hyperparameters and can distort global distances, which may lead to loss of high-dimensional expression information and affect the quality and stability of the constructed expression graph.
3. The paper only indirectly evaluates the effectiveness of Spa-CSMOTE through improvements in final clustering metrics. It is recommended to add direct quantitative or qualitative evaluation of the generated samples themselves.
4. The computational complexity of the method is insufficiently analyzed, which may limit its practical application. This is an important consideration for real-world usage.
5. The evaluation metrics are insufficient. It is recommended to include a broader set of metrics and explanations to strengthen the persuasive power of the paper's innovations.

**Questions:**

See the  weakness.

---

> ### Author Response · Authors · 2025-11-22
> **Response to Reviewer iug6 (Part 1/2)**
>
> Thanks for your valuable comments! Our responses are as follows:
>
> > ### **Robustness to noise in the pseudo-labels (W1)**
>
> **A1:**  **spClust is robust and pseudo-labels have only a slight impact**. The initialized pseudo-labels are related to the imbalance judgment and Spa-CSMOTE oversampling directly. However, *although the so-called pseudo-labels are "fake" in form, the data distribution they represent is real*; **our method is data-driven rather than label-dependent**, and we *enhance the robustness of spClust through periodic updates of pseudo-labels and prototypes*. We verified the robustness of spClust through extensive experiments:
>
> 1) Pseudo-labels initialized by different clustering methods on the DKD\_kidney\_000: Results show that spClust outperforms Leiden/SpatialLeiden/KMeans/Ward; though it exhibits certain fluctuations under different initialization clustering methods, **its robustness is superior to other methods**.
>
> **Table R1**: Comparison of different pseudo-label initialization methods on the DKD\_kidney\_000.
>
> |Method|Metric|Leiden|SpatialLeiden|KMeans|Ward| STDEV|
> |-|-|-|-|-|-|-|
> | **Initial**| **ACC** | 0.905| 0.715| 0.742| 0.814| 0.085|
> || **ARI** | 0.801| 0.483| 0.654| 0.762| 0.142|
> || **NMI** | 0.744| 0.563| 0.582| 0.624| 0.081|
> || **F1**  | 0.855| 0.718| 0.565| 0.640| 0.124|
> | **spClust (Ours)** | **ACC** | **0.912** | **0.905**| **0.895** | **0.849** | **0.028** (↑67%) |
> || **ARI** | **0.821** | **0.819**| **0.807** | **0.789** | **0.015** (↑89%) |
> || **NMI** | **0.746** | **0.729**| **0.716** | **0.690** | **0.024** (↑70%) |
> || **F1**  | **0.866** | **0.838**| **0.812** | **0.747** | **0.051** (↑59%) |
>
> 2) Pseudo-labels initialized by Leiden clustering with different random seeds: **Results demonstrate that Leiden clustering exhibits excellent robustness, and our method initialized with it maintains this property**.
>
> **Table R2**: Comparison of pseudo-labels initialized by Leiden with different random seeds on the DKD\_kidney\_000.
>
> | Method| Metric| seed=0 | seed=42 | seed=2026 |
> |-|-|-|-|-|
> | **Initial**| **ACC**| 0.906|0.905| 0.907|
> || **ARI** | 0.804 | 0.801| 0.806|
> || **NMI** | 0.746  | 0.744| 0.748|
> || **F1**  | 0.856  | 0.855| 0.856|
> | **spClust (Ours)** | **ACC** | 0.912| 0.912| 0.913|
> || **ARI** | 0.820  | 0.821| 0.822|
> || **NMI** | 0.744  | 0.746| 0.746|
> || **F1**  | 0.868  | 0.866| 0.866|
>
> 3) Pseudo-labels under extreme cases (i.e., random labels or ground-truth): Results show that under random labels (completely mismatched with the data similarity distribution), our method still enables effective clustering. Attribute this capability to GAT-based feature extraction and periodic updates of prototypes and pseudo-labels. Notably, even with ground-truth labels (perfectly matching the data), it does not yield optimal clustering performance, affirming that our method is data-driven, not label-driven. The so-called "pseudo-labels" actually encode the real data distribution.
>
> **Table R3**: Comparison of clustering results under extreme pseudo-labeling cases on the DKD\_kidney\_000.
>
> | Method| Metric| Random | Ground-Truth |
> | -| - |-|- |
> | **Initial**| **ACC** | 0.209  | 1.000|
> || **ARI** | 0.000  | 1.000|
> || **NMI** | 0.001  | 1.000|
> || **F1**  | 0.177  | 1.000|
> | **spClust (Ours)** | **ACC** | 0.687  | 0.907|
> || **ARI** | 0.499  | 0.821|
> || **NMI** | 0.600  | 0.735|
> || **F1**  | 0.477  | 0.849|
>
> 4) Pseudo-labels and prototypes correction ability of spClust: The robustness of our method depends primarily on periodic prototype and pseudo-label updates. Removing both leads to a significant performance drop. Notably, the removal of prototype updates has the most pronounced impact, while removing pseudo-label updates causes no significant performance degradation.
>
> **Table R4**: Ablation study on the periodic prototypes and pseudo-labels update strategy on the DKD\_kidney\_000.
>
> | Metric| spClust| w/o pseudo-labels update | w/o prototypes update | w/o both|
> | - |-| -| - | -|
> | **ACC** | 0.914 | 0.908| 0.770| 0.779 |
> | **ARI** | 0.823| 0.818| 0.557| 0.573 |
> | **NMI** | 0.740| 0.728| 0.549| 0.570 |
> | **F1**  | 0.870| 0.853| 0.695| 0.717 |

---

> ### Author Response · Authors · 2025-11-22
> **Response to Reviewer iug6 (Part 2/2)**
>
> > ### **Potential information loss when using UMAP (W2)**
>
> **A2:** **The UMAP-dimensionally reduced data is only used for calculating the expression similarity graph, not as the attribute of the graph**. We have supplemented comparative experiments by constructing expression KNN graphs using t-SNE, PCA, and the no-dimensional reduction scheme, respectively. Results show that the performance of reducing dimension using UMAP or t-SNE is better than using PCA or no dimension reduction, and t-SNE is the optimal choice. We have improved our method with t-SNE.
>
> **Table R5**: Comparison of building KNN graph with different dimension reduction methods on the DKD\_kidney\_000.
>
> ||UMAP|t-SNE|PCA|Nothing|
> |-|-|-|-|-|
> |**ACC**|0.912|0.914|0.585|0.909|
> |**ARI**|0.822|0.823|0.376|0.817|
> |**NMI**|0.743|0.740|0.360|0.734|
> |**F1**|0.864|0.870|0.509|0.860|
>
> > ### **Intuitive demonstration of Spa-CSMOTE (W3)**
>
> **A3:** We have supplemented the visualization results of synthetic cells generated by Spa-CSMOTE, focusing on the spatial distribution characteristics of the synthetic cells, which verify their biological rationality. Due to the format limitation, the images are shown in Appendix C.
>
> **Table R6**: Comparison of different spatial constraints on the DKD\_kidney\_000.
>
> ||Eq. (4)|Eq. (3)|
> |-|-|-|
> |**ARI**|0.822|0.819|
>
> > ### **Complexity analysis (W4)**
>
> **A4:** The spClust consists of the following key steps:
>
> Leiden clustering initialization → Spa-CSMOTE oversampling → KNN graph construction → GAT-based feature extraction → loss computation. The time complexity of each step is summarized in the table below:
>
> **Table R7**: Computational complexity of each module in spClust.
>
> |Leiden|Spa-CSMOTE|KNN Graph|GAT|Contrastive|Modularity|
> |--|--|--|--|--|--|
> |$\mathcal{O}(n)$|$\mathcal{O}(\sum_{i=1}^km_i(n+m_id))$|$\mathcal{O}(n^2d)$|$\mathcal{O}(n\sum_{i=1}^3d_{i-1}di+\|\mathcal{E}\|\sum_{i=1}^{3}d_i)$|$\mathcal{O}(n^2)$|$\mathcal{O}(n^2)$|
>
> Additionally, we have supplemented the time and space costs. Due to the need to generate a large number of cells for imbalanced data, our method incurs certain memory and time costs. Compared with stDGCC, the performance gain offset by the resource consumption is acceptable.
>
> **Table R8**: Time and space costs of different methods on the DKD\_kidney\_000.
>
> |Metric|KMeans|Ward|Leiden|SpatialLeiden|scDC|SpaGCN|stDGCC|scPROTEIN|Ours|
> |-|-|-|-|-|-|-|-|-|-|
> |**Time(s)**|0.14|0.55|1.59|9.25|18.12|6.86|84.28|8.26|93.07|
> |**Peak CPU(MB)/GPU(MB)**|303/NA|273/NA|392/NA|919/NA|1608/43|1441/NA|1743/219|1783/843|9882/4869|
>
> > ### **Insufficient metrics (W5)**
>
> **A5:** We have supplemented two extra indicators, namely ACC (Accuracy) and macro-F1 score, to establish a more holistic evaluation system: ARI and NMI focus on the rationality of the partitioning, ACC intuitively reflects the overall matching degree, and macro-F1 emphasizes ensuring performance robustness under class imbalance scenarios. Our method still demonstrates a competitive performance. Due to the space limitation, **the results (324 values) are shown in Table 1 in the manuscript.**
>
> Thank you again for your valuable comments! All the additional results will be incorporated into the manuscript.

---

### Author Response · Authors · 2025-12-01
**Rebuttal Summary to Area Chair**

Dear Area Chair:

Thank you to you and the reviewers for dedicating valuable time to reviewing our submission and for your attention to our rebuttal summary. We have carefully studied all reviewer comments, provided point-by-point responses, and completed targeted revisions. As of Nov 28, no further discussions had been received from the reviewers. To facilitate your efficient evaluation, we summarize the core contributions of this paper, the reviewers’ comments, and our responses/revisions as follows:

### **I. Core Contributions**

We focus on single-cell spatial proteomics data clustering, with two key innovations: **i)** Addressing the overlooked class imbalance issue in this field, we propose the spClust method to mitigate its interference on feature learning and clustering performance via spatial constrained synthetic minority oversampling technology; **ii)** Targeting the heterogeneity between protein expression and spatial location, we design a decoupled strategy to extract these two types of features, effectively reducing the impact of cell heterogeneity on clustering.

### **II. Common Concerns and Responses**

1. **Robustness of spClust**: All four reviewers expressed concerns about the impact of initial clustering pseudo-labels on the results. It is important to clarify that pseudo-labels reflect the true distribution of data similarity, which is crucial for clustering. Furthermore, pseudo-labels are not static but are periodically updated during feature learning and clustering. **We fully verified robustness through four sets of experiments across three datasets**: pseudo-label initialization with different clustering methods; pseudo-labels initialized by Leiden with different random seeds; two extreme pseudo-label scenarios (completely random and ground-truth labels); and ablation experiments on the periodic update strategies for pseudo-labels and prototypes. **All results confirm the robustness of spClust.** (see revised Section 4.7 and Appendix D.3)

2. **Lack of complexity analysis**: All four reviewers requested the addition of computational complexity analysis. We have supplemented the computational complexity analysis and time/space cost experiments. (see revised Appendix A)

3. **Principle and intuitive effect of Spa-CSMOTE (related to Reviewers iug6, Bag2, 73Tt)**:

- Added visual comparisons between cells generated by Spa-CSMOTE and cells at random locations (see revised Appendix C) to address the concern of "indirect evaluation of effectiveness";

- The difference between Eqs. 3 and 4 lies in the distinction between random locations and locations of cells generated by Spa-CSMOTE, which is clearly explained in the paper immediately following Eq. 4;

- The original dataset already includes two extreme class imbalance scenarios (DKD\_004: 0.57\%, DKD\_013: 1.07\%, see Table 1), addressing the question of "effectiveness when minority class percentage < 1\%".

4. **Novelty (related to Reviewers 3aV3, Bag2)**: We reclarified the innovations and core contributions of this paper in the rebuttal.

### **III. Other Concerns and Responses**

**Reviewer iug6**

- **Misunderstanding**: The reviewer might assume we used UMAP-dimensionally reduced data as node attributes for the expression similarity graph. In reality, UMAP results are used only to compute K-nearest neighbors, not as node attributes, and this approach outperforms calculations using raw data.

- **Insufficient metrics**: We added 2 new metrics to complement the commonly used clustering metrics (see revised Table 1).

**Reviewer 73Tt**

- **Lack of hyperparameter analysis**: We added relevant experiments as requested (see revised Appendix D.2);

- **Other scenarios**: We added two more challenging scenarios (breast cancer with 3 sections and melanoma with 3 sections). Experimental results show our method still outperforms baseline methods on these datasets (see revised Table 1).

### **IV. Clarification**

Reviewer Bag2 (confidence: 5) incorrectly categorized our task as *spatial transcriptomics clustering*. In fact, this paper focuses on *spatial proteomics clustering*, and there are significant differences between the two in terms of data characteristics and biological significance. The remaining reviewers are not absolutely certain about their assessments.

All reviewers' suggestions and comments have been fully incorporated into the revised manuscript. Thank you again for your hard work!

Sincerely,

The Authors

---

### Meta-Review · Area_Chair_Bvzz · 2026-01-12

**Summary:**

All reviewers concern the pseudo-labels quality and insufficiently analyzed computational analysis. Besides, Reviewer iug6 concerns its   insufficient evaluation metrics, Reviewer Bag2 concerns its limited novelty and unclear formular description, and Reviewer 73Tt concerns its missing hyperparameter sensitivity analysis.

**Reviewer Concerns:**

After rebuttal, the influence of pseudo-labels quality has been explored. However, it seems that the computational complexity analysis is still too concise. The claimed novelty and contributions of the work still appear somewhat limited.

**Reviewer Scores:**

After rebuttal, the newly-added evaluation metrics further illustrate the effectiveness of the proposed method,  and Reviewer iug6  may raise the score from 4 to 6. Reviewer 3aV3 may maintain the original score since he/she holds that the technical novelty or theoretical contribution is not clearly articulated.  Reviewer Bag2  may maintain the original score since he/she still concerns the augmentation effectiveness when encountering pseudo-labels with low quality. Due to larger-scale application problem, Reviewer 73Tt may maintain the original score.

---

### Decision · Program_Chairs · 2026-01-26

Reject